# Learning Noisy Halfspaces with a Margin: Massart is No Harder than Random

**Gautam Chandrasekaran**[*]
gautamc@cs.utexas.edu
UT Austin

**Vasilis Kontonis**[†]
vasilis@cs.utexas.edu
UT Austin

**Konstantinos Stavropoulos**[‡]
kstavrop@cs.utexas.edu
UT Austin

**Kevin Tian**
kjtian@cs.utexas.edu
UT Austin

## Abstract

We study the problem of PAC learning $\gamma$-margin halfspaces with Massart noise. We propose a simple proper learning algorithm, the Perspectron, that has sample complexity $\widetilde{O}((\epsilon\gamma)^{-2})$ and achieves classification error at most $\eta + \epsilon$ where $\eta$ is the Massart noise rate. Prior works [DGT19b, CKMY20] came with worse sample complexity guarantees (in both $\epsilon$ and $\gamma$) or could only handle random classification noise [DDK+23, KIT+23] — a much milder noise assumption. We also show that our results extend to the more challenging setting of learning generalized linear models with a known link function under Massart noise, achieving a similar sample complexity to the halfspace case. This significantly improves upon the prior state-of-the-art in this setting due to [CKMY20], who introduced this model.

## 1 Introduction

We study the problem of learning halfspaces with a margin, one of the oldest problems in the field of machine learning dating to work of Rosenblatt [Ros58]. Specifically, we consider the following formulation of this problem, where the label distribution is corrupted by Massart noise [MN06], where we use the following notation for halfspace hypotheses $h_{\mathbf{w}} : \mathbb{R}^d \to \{\pm 1\}$:

$$h_{\mathbf{w}}(\mathbf{x}) := \text{sign}\left(\mathbf{w} \cdot \mathbf{x}\right), \text{ for } \mathbf{w} \in \mathbb{R}^d. \tag{1}$$

**Definition 1** (Massart halfspace model). *Let $\eta \in [0, \frac{1}{2}]$ and $\gamma \in (0, 1)$. We say that a distribution $D$ on $\mathbb{B}^d \times \{\pm 1\}$ is an instance of the $\eta$-Massart halfspace model with margin $\gamma$ if the following hold.*

- *There exists $\mathbf{w}^\star \in \mathbb{R}^d$ such that $\|\mathbf{w}^\star\| = 1$,[4] and $D_{\mathbf{x}}$ has margin $\gamma$ with respect to $\mathbf{w}^\star$,[5] i.e., $\mathbf{Pr}_{\mathbf{x} \sim D_{\mathbf{x}}}[|\mathbf{w}^* \cdot \mathbf{x}| < \gamma] = 0$.*

- *For all $\mathbf{x} \in \text{supp}(D_{\mathbf{x}})$, there is an $\eta(\mathbf{x}) \in [0, \eta]$ such that $\mathbf{Pr}[y \neq h_{\mathbf{w}^\star}(\mathbf{x}) \mid \mathbf{x}] = \eta(\mathbf{x})$.*

We note that Definition 1 extends straightforwardly to general halfspaces up to rescaling (i.e., with larger domain size bounds and constant shift terms), as discussed in Remark 2.

---

[*]Supported by the NSF AI Institute for Foundations of Machine Learning (IFML).

[†]Supported by the NSF AI Institute for Foundations of Machine Learning (IFML).

[‡]Supported by the NSF AI Institute for Foundations of Machine Learning (IFML) and by scholarships from Bodossaki Foundation and Leventis Foundation.

[4]This normalization is made for convenience, as the noise assumption of Definition 1 is scale-invariant.

[5]See Section 2 for our notation; $D_{\mathbf{x}}$ is the $\mathbf{x}$-marginal of $D$, and $D_y(\mathbf{x})$ is the conditional marginal of $y \mid \mathbf{x}$.

The *Massart noise* model of Definition 1 for halfspaces (and more generally, binary classification problems) has garnered interest from the statistics, machine learning, and algorithms communities for a variety of reasons. This noise model was originally introduced as an intermediate noise model, between the simpler (from an algorithmic design standpoint) *random classification noise* (RCN) model [AL88], and the more challenging *agnostic* model [Hau92b, KSS94]. In the RCN model, $\eta(\mathbf{x})$ in Definition 1 is restricted to be $\eta$ pointwise, i.e., the noise level is uniform; polynomial-time PAC learning has long since known to be tractable under RCN [Byl94, BFKV98]. On the other hand, in the agnostic model (where learning is computationally intractable under well-studied conjectures [GR06, FGKP06, Dan16]), an adversary is allowed to arbitrarily modify an $\eta$ fraction of labels. As observed by [Slo88], the Massart noise model of Definition 1 is equivalent to allowing an *oblivious* adversary control an $\eta$ fraction of labels, where the $\eta$ fraction is crucially sampled independently at random. It was stated as a longstanding open question [Coh97, Blu03] whether this obliviousness of the adversary impacts the polynomial-time tractability of learning halfspaces, even with a margin.

For additional motivation, it is reasonable to consider Massart noise to be a more realistic model of real-life noise (even when benign) when compared to the RCN model, as it allows for some amount of non-uniformity. This made Definition 1 a possibly tractable way to relax the noise assumption, without running into the aforementioned computational barriers for agnostic learning. In a series of recent exciting developments, in large part spurred by the breakthrough work of [DGT19b] who gave an (improper) polynomial-time PAC learning algorithm in the Massart halfspace model, significant algorithmic advances have been made towards understanding the polynomial-time tractability of learning under Massart noise [ABHU15, DGT19b, DKTZ20a, CKMY20, ZL21, DKK$^+$22]. However, less is understood about the fine-grained sample and computational complexity of these problems, which is potentially of greater interest from a practical perspective.

We investigate this question of fine-grained complexity for the Massart halfspace model, inspired by a line of recent work on *semi-random models* [BS95], a popular framework for understanding the overfitting of algorithms to their modeling assumptions. To motivate semi-random models, observe that from a purely information-theoretic standpoint, one might suspect that learning under Massart noise is actually *easier* than RCN; the noise level $\eta(\mathbf{x})$ is only allowed to decrease, giving more "signal" with respect to $\mathbf{w}^\star$. However, this modification poses challenges when designing algorithms, e.g., because it breaks independence between $y$ and $\mathbf{x}$ beyond the value $\mathrm{sign}(\mathbf{w}^\star \cdot \mathbf{x})$. Indeed, this is reflected in our current knowledge of halfspace learning algorithms. While it is known that one can learn halfspaces with margin $\gamma$ under the RCN model to $\epsilon$ error (in the zero-one loss) using $\widetilde{O}((\epsilon\gamma)^{-2})$ samples [DDK$^+$23, KIT$^+$23], state-of-the-art learners under Massart noise use $\widetilde{O}(\gamma^{-4}\epsilon^{-3})$ samples (if required to be proper) [CKMY20] or $\widetilde{O}(\min(\gamma^{-4}\epsilon^{-3}, \gamma^{-3}\epsilon^{-5}))$ samples (otherwise) [DGT19b]. The semi-random model framework posits that this discrepancy reflects a lack of robustness in the current algorithmic theory for learning halfspaces, due to their overfitting to the RCN assumption.

For many statistical learning problems, new algorithms have been developed under semi-random modeling assumptions, with guarantees matching, or nearly-matching, classical algorithms under the corresponding fully random models [CG18, KLL$^+$23, JLM$^+$23, GC23, BGL$^+$24]. This leads us to our motivating problem, which aims to accomplish this goal for learning halfspaces.

> *Is it possible to design algorithms for learning in the Massart halfspace model with sample complexities matching the state-of-the-art for learning in the RCN model?* (2)

## 1.1 Our results

As our main contribution, we resolve (2) in the affirmative in the setting of Definition 1. We also extend our results a substantial generalization of this model in Definition 2.

**Massart halfspace model.** We begin with our basic result in the setting of the Massart halfspace model, Definition 1. Our goal in this setting is to find a *proper* hypothesis halfspace $h_{\mathbf{w}}(\mathbf{x}) = \mathrm{sign}(\mathbf{w} \cdot \mathbf{x})$ for $\mathbf{w} \in \mathbb{B}^d$, achieving good zero-one loss $\ell_{0\text{-}1}$ (see Section 2 for a definition) over examples $(\mathbf{x}, y)$ drawn from the distribution $D$. Our main result to this end is the following.

**Theorem 1** (Informal, see Theorem 3). *Let $D$ be an instance of the $\eta$-Massart halfspace model with margin $\gamma$, and let $\epsilon \in (0, 1)$. Then, Perspectron (Algorithm 1) returns $\mathbf{w} \in \mathbb{B}^d$ such that $\ell_{0\text{-}1}(\mathbf{w}) \le \eta + \epsilon$ with probability $0.99$,[6] using $\widetilde{O}(\gamma^{-2}\epsilon^{-2})$ samples and $\widetilde{O}(d\gamma^{-2}\epsilon^{-4})$ time.*

We pause to comment on Theorem 1. First, our error guarantee is of the form $\eta + \epsilon$ rather than the more stringent goal of $\ell_{0\text{-}1}(\mathbf{w}^\star) + \epsilon = \mathbf{E}_{\mathbf{x}\sim D_{\mathbf{x}}}[\eta(\mathbf{x})] + \epsilon$. There is strong evidence that this distinction is necessary for polynomial-time algorithms in the statistical query (SQ) framework of [Kea98], which our algorithm is an instance of, due to [CKMY20, DK20, NT22]. Next, the sample complexity bound of Theorem 1 matches the results of [DDK+23, KIT+23], the state-of-the-art under the milder RCN model. There is evidence that the dependences in Theorem 1 on both $\epsilon^{-1}$ and $\gamma^{-1}$ are individually tight. In particular, [MN06] shows the sample complexity of the problem is $\widetilde{\Omega}(\gamma^{-2}\epsilon^{-1})$, and [DDK+23] shows any efficient algorithm in the SQ framework must use $\widetilde{\Omega}(\gamma^{-1/2}\epsilon^{-2})$ samples. We also remark that we can assume without loss of generality that $\eta$ is known (see Appendix B.1).

Finally, as mentioned, prior to our work, the best-known polynomial-time learners under Definition 1 had sample complexities $\widetilde{O}(\min(\gamma^{-4}\epsilon^{-3}, \gamma^{-3}\epsilon^{-5}))$ [CKMY20, DGT19b]. In Table 1, we summarize relevant sample complexity bounds for learning variants of halfspace models with noise.

| Source | RCN | Massart | Proper | Sample Complexity |
|:---:|:---:|:---:|:---:|:---:|
| [DGT19b] | ✓ | ✗ | ✓ | $\gamma^{-4}\epsilon^{-2}$ |
| [DDK+23, KIT+23] | ✓ | ✗ | ✓ | $\gamma^{-2}\epsilon^{-2}$ |
| [DGT19b] | ✓ | ✓ | ✗ | $\gamma^{-3}\epsilon^{-5}$ |
| [CKMY20] | ✓ | ✓ | ✓ | $\gamma^{-4}\epsilon^{-3}$ |
| **Theorem 1** | ✓ | ✓ | ✓ | $\gamma^{-2}\epsilon^{-2}$ |

Table 1: Sample complexities of learning halfspaces with $\gamma$ margin, omitting logarithmic factors and failure probabilities for brevity. All the algorithms above run in polynomial time.

**Massart generalized linear models.** Our second result is an extension of Theorem 1 to the more challenging setting of learning generalized linear models (GLMs) with a known link function $\sigma$ under Massart noise. As before, we only consider distributions that have a margin with respect to the optimal halfspace. We now formally define the setting we study.

**Definition 2** (Massart GLM). *Let $\sigma : [-1, 1] \to [-1, 1]$ be an odd, non-decreasing function. We say that a distribution $D$ on $\mathbb{B}^d \times \{\pm 1\}$ is an instance of the $\sigma$-Massart generalized linear model (GLM) with margin $\gamma$ if the following conditions hold.*

1. *There exists $\mathbf{w}^\star \in \mathbb{R}^d$ such that $\|\mathbf{w}^\star\| = 1$ and $\mathbf{Pr}[|\mathbf{w}^* \cdot \mathbf{x}| < \gamma] = 0$.*
2. *For all $\mathbf{x} \in \mathrm{supp}(D_{\mathbf{x}})$, it holds that $\eta(\mathbf{x}) := \mathbf{Pr}[y \ne h_{\mathbf{w}^\star}(\mathbf{x}) \mid \mathbf{x}] \le \frac{1 - |\sigma(\mathbf{w}^\star \cdot \mathbf{x})|}{2}$.*

**Remark 1.** *We remark that the assumption that $\sigma$ is odd is also commonly used in prior works (see, e.g., [CN08, DKTZ20a, CKMY20]). We further show that our result extends to $\sigma$ with bounded asymmetry, albeit with a weaker error guarantee (see Definition 3 and Theorem 4).*

To provide intuition for Definition 2, observe that that if $\eta(\mathbf{x}) = \frac{1 - |\sigma(\mathbf{w}^\star \cdot \mathbf{x})|}{2}$ for some $\mathbf{x} \in \mathrm{supp}(D_{\mathbf{x}})$, then $\mathbf{E}[y \mid \mathbf{x}] = |\sigma(\mathbf{w}^\star \cdot \mathbf{x})|\mathrm{sign}(\mathbf{w}^\star \cdot \mathbf{x}) = \sigma(\mathbf{w}^\star \cdot \mathbf{x})$, i.e., Definition 2 corresponds to the standard GLM definition. In Definition 2 (compared to Definition 1), we replace the fixed noise rate upper bound $\eta$ with a data-dependent upper bound which is monotone (i.e., decreases as $|\mathbf{w}^\star \cdot \mathbf{x}|$ grows more confident). That is, Definition 2 generalizes the problem of learning Massart halfspaces, which follows by taking $\sigma(t) = (1 - 2\eta)\mathrm{sign}(t)$ for all $t \in [-1, 1]$.

When working with a Massart GLM $D$, we define $\mathrm{opt}_{\mathrm{RCN}} := \mathbf{E}_{(\mathbf{x},y)\sim D}[\frac{1 - |\sigma(\mathbf{w}^\star \cdot \mathbf{x})|}{2}]$. Note that in the special case of a Massart halfspace model, we simply have $\mathrm{opt}_{\mathrm{RCN}} = \eta$. As in the case of Massart half-

---

[6]The formal variant, Theorem 3, gives high-probability bounds at a mild polylogarithmic overhead in sample and runtime complexities.

spaces, known SQ lower bounds make competing with $\mathrm{opt} := \ell_{0\text{-}1}(\mathbf{w}^\star) := \mathbf{Pr}_{(\mathbf{x},y)\sim D}\left[y \neq h_{\mathbf{w}^\star}(x)\right]$ an intractable target, so our focus is again on attaining $\ell_{0\text{-}1}(\mathbf{w}) \approx \mathrm{opt}_{\mathrm{RCN}}$.

To our knowledge, the model in Definition 2 was first studied in [CKMY20], though we note that similar models have been considered in prior works [ZLC17, HKLM20, DKTZ20a], which we describe and compare to Definition 2 in Section 1.3. In [CKMY20], the parameterization of this model is slightly different; they assume $\sigma$ is $L$-Lipschitz and that $\mathbf{Pr}_{\mathbf{x}\sim D_{\mathbf{x}}}[|\sigma(\mathbf{w}^* \cdot \mathbf{x})| \geq \gamma] = 1$, i.e., they impose a margin on $\sigma(\mathbf{w}^* \cdot \mathbf{x})$ rather than $\mathbf{w}^* \cdot \mathbf{x}$. This implies our margin assumption (with $\gamma \leftarrow \frac{\gamma}{L}$ in Definition 2), but not vice versa. Under their slightly more restrictive assumptions, [CKMY20] claims a runtime which is an unspecified polynomial in $L\gamma^{-1}\epsilon^{-1}$, that is at least $\widetilde{\Omega}(L^4\gamma^{-4}\epsilon^{-6})$ when specialized to the halfspace case (see their Theorems 5.2 and 6.14). On the other hand, we achieve improved rates extending our simple algorithmic approach for the Massart halfspace case in this more challenging setting.

**Theorem 2** (Informal, see Theorem 4). *Let $D$ be an instance of the $\sigma$-Massart GLM with margin $\gamma$, and let $\epsilon \in (0,1)$. There is an algorithm returning $\mathbf{w} \in \mathbb{B}^d$ so that $\ell_{0\text{-}1}(\mathbf{w}) \leq \mathrm{opt}_{\mathrm{RCN}} + \epsilon$ with probability $0.99$, using $\widetilde{O}(\gamma^{-2}\epsilon^{-4})$ samples and $\widetilde{O}(d\gamma^{-2}\epsilon^{-6})$ time.*

In particular, parameterizing our problem using the margin and Lipschitz assumptions in [CKMY20] (with $\gamma \leftarrow \frac{\gamma}{L}$), we obtain an improved sample complexity of $\widetilde{O}(L^2\gamma^{-2}\epsilon^{-4})$.

## 1.2 Technical overview

**Learning Massart halfspaces.** Our learning algorithms are inspired by the certificate framework for learning with semi-random noise developed in [DKTZ20d, CKMY20]. In that framework, given a sub-optimal hypothesis $\mathbf{w}$, i.e., with error $\mathbf{Pr}_{(\mathbf{x},y)\sim D}[\mathrm{sign}(\mathbf{w} \cdot \mathbf{x}) \neq y] \geq \eta + \epsilon$, the goal is to construct a certificate of sub-optimality in the form of a separating hyperplane between $\mathbf{w}$ and the target $\mathbf{w}^*$, i.e., a vector $\mathbf{g}$ such that $\mathbf{g} \cdot \mathbf{w} \geq \mathbf{g} \cdot \mathbf{w}^\star \iff \mathbf{g} \cdot (\mathbf{w} - \mathbf{w}^\star) \geq 0$. Given such a separating hyperplane, prior works rely on cutting-plane methods (e.g., [Vai96]) or on first-order regret minimization methods to learn a hypothesis achieving the target error.

We first describe our algorithm in the halfspace setting, by motivating our choice of a certificate. Prior work [CKMY20] achieving a proper Massart halfspace learner uses the gradient of the Leaky-ReLU objective $\ell_\eta(t) := (1-\eta)\max(0,t) - \eta\max(0,-t)$ conditioned on a band around the current hypothesis $\mathbf{w}$ as a separating hyperplane. That is, they argue that for some appropriate interval $I$, it holds that $\mathbf{E}[\nabla\ell_\eta(-y\mathbf{w} \cdot \mathbf{x}) \mid \mathbf{x} \cdot \mathbf{w} \in I] \cdot (\mathbf{w} - \mathbf{w}^*) \geq 0$. This yields a sample complexity scaling as $\tilde{O}(\epsilon^{-3})$ because one has to sample condionally from the band $I$ and estimate the gradient of the Leaky-ReLU on these samples up to error $\epsilon$, as well as additional overhead in $\gamma^{-1}$ due to use of expensive outer loops taking advantage of these certificates, such as cutting-plane methods.

To avoid the sample complexity overhead of this conditioning (implemented via rejection sampling), we use a simple reweighting scheme pointwise, which puts a larger weight of $|\mathbf{w} \cdot \mathbf{x}|^{-1}$ (an inverse margin) on points closer to the boundary of our current hypothesis $h_{\mathbf{w}}$. Intuitively, this reweighting is a soft implementation of the hard conditioning done in [CKMY20]. This is motivated by our first important observation: any significantly sub-optimal hypothesis $\mathbf{w}$ with $\ell_{0\text{-}1}(\mathbf{w}) \geq \eta + \epsilon$ satisfies

$$\mathbf{E}\left[\frac{\nabla\ell_\eta(-y\mathbf{w} \cdot \mathbf{x})}{|\mathbf{w} \cdot \mathbf{x}|}\right] \cdot (\mathbf{w} - \mathbf{w}^\star) \geq \epsilon,$$

proven in Lemma 1. This suggests using $\mathbf{g} = \mathbf{E}[\nabla\ell_\eta(-y\mathbf{w} \cdot \mathbf{x})|\mathbf{w} \cdot \mathbf{x}|^{-1}]$ (rather than $\mathbf{E}[\nabla\ell_\eta(-y\mathbf{w} \cdot \mathbf{x}) \mid \mathbf{x} \cdot \mathbf{w} \in I]$ as in [CKMY20]) as our certificate, which we can estimate via a single sample.

While reweighting by the inverse margin $|\mathbf{w} \cdot \mathbf{x}|^{-1}$ gives a separating hyperplane certificate, it may be impossible to estimate this certificate from few samples, e.g., if the weight $|\mathbf{w} \cdot \mathbf{x}|^{-1}$ is often very large, which introduces significant variance. For instance, even if $D_{\mathbf{x}}$ has margin with respect to a target $\mathbf{w}^\star$, this is not necessarily true with respect to the current hypothesis $\mathbf{w}$ (without further distributional assumptions on $D_{\mathbf{x}}$). To overcome this, we change our pointwise reweighting to be less aggressive and instead use $(|\mathbf{w} \cdot \mathbf{x}| + \gamma)^{-1}$. In Lemma 2, we exploit the margin assumption about $D_{\mathbf{x}}$ to show that when $\ell_{0\text{-}1}(\mathbf{w}) \geq \eta + \epsilon$, it is still the case that $\mathbf{E}[\nabla\ell_\eta(-y\mathbf{w} \cdot \mathbf{x})(|\mathbf{w} \cdot \mathbf{x}| + \gamma)^{-1}] \cdot (\mathbf{w} - \mathbf{w}^*) \geq \epsilon$. Moreover, we still have an unbiased estimator for this separating hyperplane $\mathbf{E}[\nabla\ell_\eta(-y\mathbf{w} \cdot \mathbf{x})(|\mathbf{w} \cdot \mathbf{x}| + \gamma)^{-1}]$, and the estimator is bounded in Euclidean norm by $\gamma^{-1}$ with probability 1 by our margin assumption.

Standard concentration inequalities now show $\widetilde{O}(\gamma^{-2}\epsilon^{-2})$ samples suffice to obtain a separation oracle with high probability. Plugging this certificate into oracle-efficient cutting-plane methods (e.g., [Vai96]) implies an algorithm with sample complexity $\widetilde{O}(\gamma^{-4}\epsilon^{-2})$ (after a random projection process [AV99] to reduce to $\widetilde{O}(\gamma^{-2})$ dimensions). This already improves upon the prior best-known $\widetilde{O}(\gamma^{-4}\epsilon^{-3})$ sample complexity. We further improve our dependence on $\gamma$ by using it in a perceptron-like regret minimization scheme, where at every step we update the current hypothesis $\mathbf{w}$ using the aforementioned bounded unbiased estimator of our certificate, see Lemma 3 and Algorithm 1. Overall, our algorithm iterates the following *very simple update* for a step-size $\lambda > 0$ and $\beta := 1 - 2\eta$:

$$\mathbf{w}^{(t+1)} \leftarrow \mathbf{w}^{(t)} - \lambda(\beta\mathrm{sign}(\mathbf{w}^{(t)} \cdot \mathbf{x}^{(t)}) - y^{(t)})\frac{\mathbf{x}^{(t)}}{|\mathbf{w}^{(t)} \cdot \mathbf{x}^{(t)}| + \gamma} \quad \text{with} \quad \mathbf{w}^{(0)} \leftarrow \mathbf{0}. \tag{3}$$

Since at every step we perform an (approximate) perspective projection $(|\mathbf{w} \cdot \mathbf{x}| + \gamma)^{-1}$ on our sample, we call Algorithm 1 which iterates the update in Equation (3) the Perspectron.

**Learning Massart GLMs.** For learning Massart GLMs (Definition 2), we use a similar certificate-based approach. While it is simple to show reweighting with the inverse margin $|\mathbf{w} \cdot \mathbf{x}|^{-1}$ still works in this case (see Lemma 5), using the bounded reweighting $(|\mathbf{w} \cdot \mathbf{x}| + \gamma)^{-1}$ does not. Instead we use a new reweighting of of the form $(|\mathbf{w} \cdot \mathbf{x}| + \alpha\gamma)$ where $\alpha = O(\epsilon)$ (see Lemma 6). Using a similar iterative method as the Perspectron defined in (3), we obtain our sample complexity of $\widetilde{O}(\gamma^{-2}\epsilon^{-4})$ for learning Massart GLMs.

## 1.3   Related work

We briefly survey some additional related works here. First, a common worst-case assumption used in statistical learning is that the label noise is adversarial (a.k.a. agnostic) [Hau92a]. In that setting, a lot of progress has been made for learning halfspaces when the underlying distribution satisfies structural assumptions (e.g., it is Gaussian or log-concave) [KKMS05, KOS08, ABHU15, DKS18, DKTZ20c, DKTZ22]. For learning halfspaces with a margin, the best-known agnostic results have runtime and sample complexity that depend exponentially on the margin $\gamma$ and/or the accuracy parameter $\epsilon$ [SSS09, LS11, DKM19]. Another important line of work [DKTZ20d, DKK$^+$20, ZL21] has focused on learning halfspaces under Tsybakov noise: a semi-random noise model that extends Massart noise, but is still easier than the agnostic setting. We also note that variants of Definition 2 have appeared before: the *generalized Tsybakov low noise condition* of [HKLM20] is a close relative which imposes different noise rates within and outside a margin, and the *strong Massart noise* of [ZLC17, DKTZ20a] is an instance of Definition 2 without the margin restriction.

Our algorithms rely on the certificate framework developed in [DKTZ20b, CKMY20] and the Leaky-ReLU loss that has been extensively used in prior works on learning with random classification and Massart label noise [Byl98, DGT19a, CKMY20, DKT21]. Our main technical contribution is a new certificate that relies on an inverse-margin reweighting scheme and can be estimated using a single sample at every iteration. Similar, "inverse-margin" reweighting schemes have been used for learning general halfspaces [CKMY20] and online linear classification [DKTZ24]. Those results have no implications for the sample complexity of the problem studied here. Finally, we mention that a local reweighting scheme that is somewhat similar in spirit to ours (but very different in its implementation) was previously employed by [KLL$^+$23], for a different semi-random statistical learning problem.

## 1.4   Limitations and open problems

One interesting open direction is providing improved sample complexity guarantees for more general noise models. For example, the misspecified GLM framework (Definition 3.2, [CKMY20]) generalizes Definition 2 to include an additional misspecification parameter $\zeta$ such that $\eta(\mathbf{x}) \leq \frac{1 - |\sigma(\mathbf{w}^* \cdot \mathbf{x})|}{2} + \zeta$. Our approach does not directly apply in this setting, since $\zeta = 0$ is important for our separation oracle result of Lemma 6. A different interesting generalization of the noise model corresponds to the case where the link function $\sigma$ is unknown, which the [CKMY20] algorithm can handle (at a much higher sample complexity). While our algorithms require knowledge of $\sigma$, it would be interesting to explore whether techniques from learning single-index models (e.g., [KS09, GGKS23, ZWDD24]) can be used to extend our algorithms in this setting.

A clear next step is to design efficient algorithms with sample complexities independent of the margin but still linear in the dimension $d$.[7], e.g., with an $\approx d\epsilon^{-2}$ sample complexity. In prior work [DKT21] such an algorithm is given, albeit with a significantly worse $\mathrm{poly}(d\epsilon^{-1})$ sample complexity.

## 2 Preliminaries

We denote vectors in lower-case boldface, and $\|\mathbf{x}\|$ is the Euclidean norm of $\mathbf{x} \in \mathbb{R}^d$. We use $\mathbb{B}^d$ to denote the unit ball in $\mathbb{R}^d$, i.e. $\mathbb{B}^d := \{\mathbf{x} \in \mathbb{R}^d \mid \|\mathbf{x}\| \leq 1\}$. We use $\mathbf{\Pi}_{\mathbb{B}^d}(\mathbf{w}) := \min\{1, \frac{1}{\|\mathbf{w}\|}\}\mathbf{w}$ to denote the Euclidean projection of a vector $\mathbf{w} \in \mathbb{R}^d$ onto $\mathbb{B}^d$. We let $\mathbb{0}_d$ be the all-zeroes vector in dimension $d$. We reserve the overline notation $\bar{\mathbf{x}}$ to denote the unit vector in the direction of $\mathbf{x}$, i.e. $\bar{\mathbf{x}} := \frac{\mathbf{x}}{\|\mathbf{x}\|}$. We let $\mathrm{sign} : \mathbb{R} \to \{\pm 1\}$ be defined so $\mathrm{sign}(t) = 1$ iff $t \geq 0$. We use $\mathbb{1}\{\mathcal{E}\}$ to denote the 0-1 indicator of a random event $\mathcal{E}$, $\mathbf{Pr}[\mathcal{E}]$ to denote its probability, and $\mathbf{E}$ to denote the expectation operator. The support of a distribution $D$ is denoted $\mathrm{supp}(D)$, and $[N] := \{i \in \mathbb{N} \mid i \leq N\}$.

Throughout the paper we study the problem of learning a binary classifier, given labeled examples from a distribution $D$ over labeled examples $(\mathbf{x}, y) \in \mathbb{R}^d \times \{0, 1\}$, under various models on the distribution to be discussed. We refer to the $\mathbf{x}$-marginal of $D$ by $D_{\mathbf{x}}$, and the conditional distribution of the label $y \mid \mathbf{x}$ by $D_y(\mathbf{x})$. We will primarily be interested in learning halfspace hypotheses, which for $\mathbf{w} \in \mathbb{R}^d$ are the corresponding functions $h_{\mathbf{w}} : \mathbb{R}^d \to \{\pm 1\}$ defined by $h_{\mathbf{w}}(\mathbf{x}) := \mathrm{sign}(\mathbf{w} \cdot \mathbf{x})$. We also denote the *zero-one loss* of a halfspace hypothesis $h_{\mathbf{w}}$ corresponding to $\mathbf{w} \in \mathbb{R}^d$ as follows, when the distribution $D$ over labeled examples is clear from context: $\ell_{0\text{-}1}(\mathbf{w}) := \mathbf{Pr}_{(\mathbf{x},y) \sim D}[h_{\mathbf{w}}(\mathbf{x}) \neq y]$. We define the Leaky-ReLU function with parameter $\lambda > 0$ as $\ell_\lambda(t) := (1 - \lambda)\max(0, t) - \lambda \max(0, -t)$. Given vectors $\mathbf{w}, \mathbf{x}$ and $y \in \{\pm 1\}$, it holds that the (sub) gradient of $\ell_\lambda(-y\mathbf{w} \cdot \mathbf{x})$ with respect to $\mathbf{w}$ is $\nabla_{\mathbf{w}}\ell_\lambda(-y\mathbf{w} \cdot \mathbf{x}) = \frac{1}{2}((1 - 2\lambda)\mathrm{sign}(\mathbf{w} \cdot \mathbf{x}) - y) \cdot \mathbf{x}$. We also provide some brief remarks on how to extend Definition 1 to more general settings here.

**Remark 2.** *Definition 1 extends straightforwardly to the case where $D_{\mathbf{x}}$ has margin $\gamma$ and is supported on a subset of $R \cdot \mathbb{B}^d$ for $R \neq 1$, by rescaling so $R \leftarrow 1$ and $\gamma \leftarrow \frac{\gamma}{R}$, as halfspace hypotheses and our label noise assumptions only depend on signs. Other than these margin and support assumptions, we make no additional distributional assumptions about the $\mathbf{x}$-marginal $D_{\mathbf{x}}$. Further, due to working in the distributional assumption-free setting, we can assume with up to constant factor loss(in margin) that the halfspace is homogeneous, i.e., has no constant shift term. That is, given a halfspace $\mathrm{sign}(\mathbf{w} \cdot \mathbf{x} + b)$ with $\|\mathbf{w}\|, |b| \leq 1$, after a feature expansion ($e : \mathbf{x} \mapsto \frac{1}{\sqrt{2}}(\mathbf{x}, 1)$) the halfspace $h_{\mathbf{w}'}(\mathbf{z}) = \mathrm{sign}((\mathbf{w}, b) \cdot \mathbf{z})$ is homogeneous while still having a margin $\geq \frac{\gamma}{2}$ with respect to $\mathbf{w}' = \frac{1}{\sqrt{1+b^2}}(\mathbf{w}, b)$. Finally, the Massart noise model of [MN06] is defined for any hypothesis class and is not tied to halfspaces. Since we focus on learning halfspaces with a margin, we combined the hypothesis class $\{h_{\mathbf{w}}\}_{\mathbf{w} \in \mathbb{R}^d}$ with the noise model in Definition 1 for simplicity.*

## 3 Massart halfspaces

In this section, we give our result on learning Massart halfspaces with margin. Our proof is surprisingly short, and we separate it into its two main components: a structural lemma in Section 3.1 which shows how to estimate a separating hyperplane given a sub-optimal $\mathbf{w}$, and a perceptron-like analysis of a stochastic iterative method in Section 3.2.

### 3.1 Separating hyperplanes for Massart halfspaces

We prove our main structural lemma here, used to argue the progress of our iterative method. As highlighted in Section 1.2, we show that when the current $\ell_{0\text{-}1}(\mathbf{w}) \geq \eta + \epsilon$, we can construct an unbiased estimator for a separating hyperplane between $\mathbf{w}$ and the target vector $\mathbf{w}^\star$.

**Warmup: an "unbounded" separating hyperplane.** Before presenting the full proof, we first give a separating hyperplane that works for any feature distribution — even without margin assumptions. The proposed separating hyperplane works due to the fact that we can express the zero-one in terms of the Leaky-ReLU which is a convex function, as was previously observed by [DGT19b].

---

[7]By standard random-projection procedures [AV99], the dimension $d$ is comparable to $\gamma^{-2}$ under a $\gamma$-margin assumption, and therefore our sample complexity is nearly-linear in the "dimension."

**Lemma 1** (Separating hyperplane for Massart halfspaces). *Let $D$ be an instance of the $\eta$-Massart halfspace model, and $\mathbf{w} \in \mathbb{R}^d$ has classification error $\ell_{0\text{-}1}(\mathbf{w}) \geq \eta + \epsilon$. It holds that* $\mathbf{E}_{(\mathbf{x},y) \sim D}\left[\frac{\nabla_{\mathbf{w}}\ell_{\eta}(-y\mathbf{w} \cdot \mathbf{x})}{|\mathbf{w} \cdot \mathbf{x}|}\right] \cdot (\mathbf{w} - \mathbf{w}^*) \geq \epsilon$.

*Proof.* We recall Claim 2.1 from [DGT19b]: for all $\mathbf{w}, \mathbf{x}$, it holds that $\mathbf{E}_{y \sim D_y(\mathbf{x})}\left[\ell_{\lambda}(-y\mathbf{w} \cdot \mathbf{x})\right] = (\mathbf{Pr}_{y \sim D_y(\mathbf{x})}\left[\text{sign}(\mathbf{w} \cdot \mathbf{x}) \neq y\right] - \lambda) \cdot |\mathbf{w} \cdot \mathbf{x}|$. In particular, we have $\mathbf{E}_{(\mathbf{x},y) \sim D}\left[\frac{\ell_{\eta}(-y\mathbf{w} \cdot \mathbf{x})}{|\mathbf{w} \cdot \mathbf{x}|}\right] = \ell_{0\text{-}1}(\mathbf{w}) - \eta$. From the convexity of $\ell_{\eta}(-y\mathbf{w} \cdot \mathbf{x})$, we obtain that $\nabla_{\mathbf{w}}\ell_{\eta}(-y\mathbf{w} \cdot \mathbf{x}) \cdot (\mathbf{w} - \mathbf{w}^*) \geq \ell_{\eta}(-y\mathbf{w} \cdot \mathbf{x}) - \ell_{\eta}(-y\mathbf{w}^* \cdot \mathbf{x})$. By dividing both sides by $|\mathbf{w} \cdot \mathbf{x}|$ and taking expectation over $\mathbf{x}$ and $y$, we obtain

$$\mathbf{E}_{\mathbf{x},y}\left[\frac{\nabla_{\mathbf{w}}(\ell_{\eta}(-y\mathbf{w} \cdot \mathbf{x}))}{|\mathbf{w} \cdot \mathbf{x}|}\right] \cdot (\mathbf{w} - \mathbf{w}^*) \geq \mathbf{E}_{\mathbf{x},y}\left[\frac{\ell_{\eta}(-y\mathbf{w} \cdot \mathbf{x})}{|\mathbf{w} \cdot \mathbf{x}|}\right] - \mathbf{E}_{\mathbf{x},y}\left[\frac{\ell_{\eta}(-y\mathbf{w}^* \cdot \mathbf{x})}{|\mathbf{w} \cdot \mathbf{x}|}\right] \geq \epsilon,$$

where the last inequality follows from the following facts: (1) for all $\mathbf{x}$, it holds that $\mathbf{E}_{y \sim D_y(\mathbf{x})}\left[\ell_{\eta}(-y\mathbf{w}^* \cdot \mathbf{x})\right] = (\mathbf{Pr}_{y \sim D_y(\mathbf{x})}\left[\text{sign}(\mathbf{w}^* \cdot \mathbf{x}) \neq y\right] - \eta) \cdot |\mathbf{w}^* \cdot \mathbf{x}| = (\eta(\mathbf{x}) - \eta) \cdot |\mathbf{w}^* \cdot \mathbf{x}| \leq 0$, and (2) $\mathbf{E}_{\mathbf{x},y}\left[\frac{\ell_{\eta}(-y\mathbf{w} \cdot \mathbf{x})}{|\mathbf{w} \cdot \mathbf{x}|}\right] = \ell_{0\text{-}1}(\mathbf{w}) - \eta \geq \epsilon$. This completes the proof. $\quad\square$

**A "bounded" separating hyperplane for $\gamma$-margin Massart halfspaces.** Our claim in the previous lemma was very general: it works for any marginal distribution. However, as discussed in Section 1.2, the unbounded nature of this separating hyperplane may make it impossible to estimate from samples. To overcome this, we propose a new candidate hyperplane: $\mathbf{E}_{\mathbf{x},y}\left[\frac{\nabla_{\mathbf{w}}\ell_{\eta}(-y\mathbf{w} \cdot \mathbf{x})}{|\mathbf{w} \cdot \mathbf{x}| + \gamma}\right]$. We prove that this candidate is indeed a separating hyperplane by leveraging the fact that we have margin $\gamma$ with respect to the optimal halfspace $\mathbf{w}^*$. Recall from Section 2 that $\nabla_{\mathbf{w}}\ell_{\eta}(-y\mathbf{w} \cdot \mathbf{x}) = \frac{1}{2}((1 - 2\eta)\text{sign}(\mathbf{w} \cdot \mathbf{x}) - y)$.

**Lemma 2** (Bounded separating hyperplane for Massart halfspaces). *Let $D$ be an instance of the $\eta$-Massart halfspace model with margin $\gamma$ (with respect to $\mathbf{w}^*$) and define $\beta = 1 - 2\eta$. If $\mathbf{w} \in \mathbb{R}^d$ has $\ell_{0\text{-}1}(\mathbf{w}) \geq \eta + \epsilon$, it holds that*

$$\mathbf{E}_{(\mathbf{x},y) \sim D}\left[(\beta\text{sign}(\mathbf{w} \cdot \mathbf{x}) - y)\frac{\mathbf{x}}{|\mathbf{w} \cdot \mathbf{x}| + \gamma}\right] \cdot (\mathbf{w} - \mathbf{w}^*) \geq 2\epsilon.$$

*Proof.* We first observe that by the definition of the Massart halfspace model, $\mathbf{E}_{y \sim D_y(\mathbf{x})}[y] = (1 - 2\eta(\mathbf{x}))\text{sign}(\mathbf{w}^* \cdot \mathbf{x}) = \beta(\mathbf{x})\text{sign}(\mathbf{w}^* \cdot \mathbf{x})$, where $\beta(\mathbf{x}) := 1 - 2\eta(\mathbf{x})$. Therefore, we have that

$$
\begin{aligned}
I &:= \mathbf{E}_{(\mathbf{x},y) \sim D}\left[(\beta\text{sign}(\mathbf{w} \cdot \mathbf{x}) - y)\frac{(\mathbf{w} \cdot \mathbf{x} - \mathbf{w}^* \cdot \mathbf{x})}{|\mathbf{w} \cdot \mathbf{x}| + \gamma}\right] \\
&= \mathbf{E}_{\mathbf{x} \sim D_{\mathbf{x}}}\left[(\beta\text{sign}(\mathbf{w} \cdot \mathbf{x}) - \beta(\mathbf{x})\text{sign}(\mathbf{w}^* \cdot \mathbf{x}))\frac{(\mathbf{w} \cdot \mathbf{x} - \mathbf{w}^* \cdot \mathbf{x})}{|\mathbf{w} \cdot \mathbf{x}| + \gamma}\right].
\end{aligned}
$$

We denote by $g(\mathbf{x}) := (\beta\text{sign}(\mathbf{w} \cdot \mathbf{x}) - \beta(\mathbf{x})\text{sign}(\mathbf{w}^* \cdot \mathbf{x}))\frac{(\mathbf{w} \cdot \mathbf{x} - \mathbf{w}^* \cdot \mathbf{x})}{|\mathbf{w} \cdot \mathbf{x}| + \gamma}$, which we bound differently based on whether $\mathbf{x}$ falls in the agreement region $A := \left\{\mathbf{x} \in \mathbb{B}^d \mid h_{\mathbf{w}^*}(\mathbf{x}) = h_{\mathbf{w}}(\mathbf{x})\right\}$. For $\mathbf{x} \in A$,

$$
\begin{aligned}
g(\mathbf{x}) &= \big(\beta\text{sign}(\mathbf{w} \cdot \mathbf{x}) - \beta(\mathbf{x})\text{sign}(\mathbf{w}^* \cdot \mathbf{x})\big)\frac{(\mathbf{w} \cdot \mathbf{x} - \mathbf{w}^* \cdot \mathbf{x})}{|\mathbf{w} \cdot \mathbf{x}| + \gamma} \\
&= \big(\beta - \beta(\mathbf{x})\big)\frac{|\mathbf{w} \cdot \mathbf{x}| - |\mathbf{w}^* \cdot \mathbf{x}|}{|\mathbf{w} \cdot \mathbf{x}| + \gamma} \geq \beta - \beta(\mathbf{x}).
\end{aligned}
$$

The second equality follows from the fact that $\text{sign}(\mathbf{w}^* \cdot \mathbf{x}) = \text{sign}(\mathbf{w} \cdot \mathbf{x})$. The final inequality holds since $\beta - \beta(\mathbf{x}) \leq 0$ and $\frac{|\mathbf{w} \cdot \mathbf{x}| - |\mathbf{w}^* \cdot \mathbf{x}|}{|\mathbf{w} \cdot \mathbf{x}| + \gamma} \leq 1$. Similarly, for $\mathbf{x} \notin A$, an analogous calculation yields $g(\mathbf{x}) = \big(\beta + \beta(\mathbf{x})\big)\frac{|\mathbf{w} \cdot \mathbf{x}| + |\mathbf{w}^* \cdot \mathbf{x}|}{|\mathbf{w} \cdot \mathbf{x}| + \gamma} \geq \beta + \beta(\mathbf{x})$. The first equality holds because $\text{sign}(\mathbf{w}^* \cdot \mathbf{x}) \neq \text{sign}(\mathbf{w} \cdot \mathbf{x})$ and the final inequality follows since $|\mathbf{w}^* \cdot \mathbf{x}| \geq \gamma$ from the margin assumption. Thus

$$I \geq \mathbf{E}_{\mathbf{x} \sim D_{\mathbf{x}}}\left[\mathbb{1}\{\mathbf{x} \in A\}(\beta - \beta(\mathbf{x}))\right] + \mathbf{E}_{\mathbf{x} \sim D_{\mathbf{x}}}\left[\mathbb{1}\{\mathbf{x} \notin A\}(\beta + \beta(\mathbf{x}))\right]. \tag{4}$$

We will now use our lower bound on $\ell_{\text{0-1}}(\mathbf{w})$, which we relate to Equation (4). We have $\ell_{\text{0-1}}(\mathbf{w}) = \mathbf{E}_{\mathbf{x} \sim D_{\mathbf{x}}}\left[\mathbb{1}\left\{\mathbf{x} \in A\right\}\eta(\mathbf{x})\right] + \mathbf{E}_{\mathbf{x} \sim D_{\mathbf{x}}}\left[\mathbb{1}\left\{\mathbf{x} \notin A\right\}(1 - \eta(\mathbf{x}))\right] = \mathbf{E}_{\mathbf{x} \sim D_{\mathbf{x}}}\left[\mathbb{1}\left\{\mathbf{x} \notin A\right\}\beta(\mathbf{x})\right] + \mathbf{E}_{x \sim D_{\mathbf{x}}}\left[\eta(\mathbf{x})\right]$. Next, by our definition $\beta(\mathbf{x}) = 1 - 2\eta(\mathbf{x})$, rearranging and expanding we have:

$$
\begin{aligned}
\ell_{\text{0-1}}(\mathbf{w}) - \eta &= \underset{\mathbf{x} \sim D_{\mathbf{x}}}{\mathbf{E}}\left[\mathbb{1}\{\mathbf{x} \notin A\}\beta(\mathbf{x})\right] + \frac{1}{2}\underset{\mathbf{x} \sim D_{\mathbf{x}}}{\mathbf{E}}\left[\beta - \beta(\mathbf{x})\right] \\
&= \underset{\mathbf{x} \sim D_{\mathbf{x}}}{\mathbf{E}}\left[\mathbb{1}\{\mathbf{x} \notin A\}\beta(\mathbf{x})\right] + \frac{1}{2}\underset{\mathbf{x} \sim D_{\mathbf{x}}}{\mathbf{E}}\left[(\mathbb{1}\{\mathbf{x} \in A\} + \mathbb{1}\{\mathbf{x} \notin A\})(\beta - \beta(\mathbf{x}))\right] \\
&= \frac{1}{2}\underset{\mathbf{x} \sim D_{\mathbf{x}}}{\mathbf{E}}\left[\mathbb{1}\{\mathbf{x} \notin A\}(\beta(\mathbf{x}) + \beta)\right] + \frac{1}{2}\underset{\mathbf{x} \sim D_{\mathbf{x}}}{\mathbf{E}}\left[\mathbb{1}\{\mathbf{x} \in A\}(\beta - \beta(\mathbf{x}))\right].
\end{aligned}
\tag{5}
$$

We finish the proof by combining Equation (4), Equation (5), and $\ell_{\text{0-1}}(\mathbf{w}) - \eta \geq \epsilon$. $\qquad\square$

## 3.2 Perspectron

We now present and analyze Perspectron, our algorithm for learning Massart halfspaces.

---

**Algorithm 1:** Perspectron

**1 Input:** $\{\mathbf{x}^i, y^i\}_{i \in [T_1 + T_2]} \subset \mathbb{R}^d \times \{\pm 1\}$ drawn i.i.d. from $D$ in the $\eta$-Massart halfspace model with margin $\gamma$, step size $\lambda > 0$, failure probability $\delta \in (0, \frac{1}{2})$

**2** $\beta \leftarrow 1 - 2\eta$, $N \leftarrow \lceil \log_2(\frac{2}{\delta}) \rceil$, $T \leftarrow \lceil \frac{T_1}{N} \rceil$

**3** $H \leftarrow \emptyset$

**4 for** $j \in [N]$ **do**

**5** $\quad$ $\mathbf{w}^{1,j} \leftarrow \mathbb{0}_d$

**6** $\quad$ **for** $t \in [\min(T, T_1 - (j-1)T)]$ **do**

**7** $\quad\quad$ $i \leftarrow (j-1)T + t$

**8** $\quad\quad$ $\mathbf{w}^{t+1,j} \leftarrow \mathbf{w}^{t,j} - \lambda\frac{\beta\,\mathrm{sign}(\mathbf{w}^{t,j}\cdot\mathbf{x}^i) - y^i}{|\mathbf{w}^{t,j}\cdot\mathbf{x}^i| + \gamma}\mathbf{x}^i$

**9** $\quad$ **end**

**10** $\quad$ $H \leftarrow H \cup \left\{\mathbf{w}^{t,j}\right\}_{t \in [\min(T, T_1 - (j-1)T)]}$

**11 end**

**12** $S \leftarrow \{\mathbf{x}^i, y^i\}_{i = T_1 + 1}^{T_1 + T_2}$

**13** $\mathbf{w} \leftarrow \arg\min_{\mathbf{w} \in H} \mathbf{Pr}_{(\mathbf{x},y) \sim_{\text{unif.}} S}[h_{\mathbf{w}}(\mathbf{x}) \neq y]$

**14 Return:** $h_{\mathbf{w}}$

---

We begin by giving a self-contained analysis of a single loop $j \in [N]$ of Line 6 to Line 9, showing that for sufficiently large $T$, at least one iterate achieves small $\ell_{\text{0-1}}$ with constant probability.

**Lemma 3.** *Let $\{\mathbf{x}^i, y^i\}_{i \in [T]} \sim_{i.i.d.} D$, where $D$ is an instance of the $\eta$-Massart halfspace model with margin $\gamma$ with respect to $\mathbf{w}^\star$. Consider iterating, from $\mathbf{w}^1 := \mathbb{0}_d$,*

$$
\mathbf{w}^{t+1} \leftarrow \mathbf{w}^t - \lambda\frac{\beta\,\mathrm{sign}(\mathbf{w}^t \cdot \mathbf{x}^t) - y^t}{|\mathbf{w}^t \cdot \mathbf{x}^t| + \gamma}\mathbf{x}^t,
\tag{6}
$$

*for $\beta := 1 - 2\eta$, $\lambda := \frac{\gamma}{2\sqrt{T}}$. Then if $T \geq \frac{16}{\epsilon^2\gamma^2}$, $\mathbf{Pr}[\min_{t \in [T]} \ell_{\text{0-1}}(\mathbf{w}^t) \geq \eta + \frac{\epsilon}{2}] \leq \frac{1}{2}$.*

*Proof.* Throughout the proof, say $\mathbf{w} \in \mathbb{R}^d$ is *bad* iff $\ell_{\text{0-1}}(\mathbf{w}) \geq \eta + \frac{\epsilon}{2}$, and let $\mathcal{E}_t$ denote the event that all of the iterates $\{\mathbf{w}^s\}_{s \in [t]}$ updated according to (6) are bad. Define the potential function $\Phi_t := \mathbf{E}[\mathbb{1}\left\{\mathcal{E}_t\right\} \cdot \|\mathbf{w}^\star - \mathbf{w}^t\|^2]$ for all $t \in [T]$. On expanding the expression for the squared norm

and using the fact that $\mathbb{1}\{\mathcal{E}_{t+1}\} \leq \mathbb{1}\{\mathcal{E}_t\}$,

$$
\begin{aligned}
\Phi_{t+1} &\leq \mathbf{E}\left[\mathbb{1}\{\mathcal{E}_t\} \cdot \left\|\mathbf{w}^\star - \mathbf{w}^{t+1}\right\|^2\right] \\
&\leq \Phi_t + \lambda^2 \mathbf{E}\left[\left\|\frac{\beta\mathrm{sign}(\mathbf{w}^t \cdot \mathbf{x}^t) - y^t}{|\mathbf{w}^t \cdot \mathbf{x}^t| + \gamma}\mathbf{x}^t\right\|^2\right] \\
&\quad - 2\lambda \mathbf{E}\left[\mathbb{1}\{\mathcal{E}_t\} \cdot \frac{\beta\mathrm{sign}(\mathbf{w}^t \cdot \mathbf{x}^t) - y^t}{|\mathbf{w}^t \cdot \mathbf{x}^t| + \gamma}\mathbf{x}^t \cdot (\mathbf{w}^t - \mathbf{w}^\star)\right] \\
&\leq \Phi_t + \frac{4\lambda^2}{\gamma^2} - 2\lambda \mathbf{Pr}[\mathcal{E}_t]\mathbf{E}\left[\frac{\beta\mathrm{sign}(\mathbf{w}^t \cdot \mathbf{x}^t) - y^t}{|\mathbf{w}^t \cdot \mathbf{x}^t| + \gamma}\mathbf{x}^t \cdot (\mathbf{w}^t - \mathbf{w}^\star) \mid \mathcal{E}_t\right] \\
&\leq \Phi_t + \frac{4\lambda^2}{\gamma^2} - 2\lambda\epsilon\,\mathbf{Pr}[\mathcal{E}_t].
\end{aligned}
$$

Here, the third inequality used $\mathbf{x}^t \in \mathbb{B}^d$ and $|\beta\mathrm{sign}(\mathbf{w}^t \cdot \mathbf{x}^t) - y^t| \leq 2$, and the fourth applied Lemma 2. Now using that $\Phi_1 \leq \|\mathbf{w}^\star\|^2 = 1$, $\Phi_{T+1} \geq 0$, and $\mathbf{Pr}[\mathcal{E}_t] \geq \mathbf{Pr}[\mathcal{E}_T]$ for $t \in [T]$, we have $2\lambda\epsilon T\mathbf{Pr}[\mathcal{E}_T] \leq 1 + \frac{4\lambda^2 T}{\gamma^2}$. The conclusion $\mathbf{Pr}[\mathcal{E}_T] \leq \frac{1}{2}$ follows from our choices of $\lambda, T$. $\qquad\square$

We next analyze the hypothesis selection step in Line 13.

**Lemma 4** (Hypothesis selection). *Suppose there exists $\hat{\mathbf{w}} \in H$ with $\ell_{\text{0-1}}(\mathbf{w}) \leq \eta + \frac{\epsilon}{2}$. Then if $T_2 \geq \frac{8}{\epsilon^2}\log(\frac{2|H|}{\delta})$, with probability $\geq 1 - \delta$ the $\mathbf{w}$ returned by Line 13 satisfies $\ell_{\text{0-1}}(\mathbf{w}) \leq \eta + \epsilon$.*

*Proof.* Because $S$ is independent of $H$, Hoeffding's inequality and a union bound implies that $\left|\mathbf{Pr}_{(\mathbf{x},y)\sim_{\text{unif.}} S}[h_\mathbf{w}(\mathbf{x}) \neq y] - \ell_{\text{0-1}}(\mathbf{w})\right| \leq \frac{\epsilon}{4}$ with probability $\geq 1 - \delta$, for all $\mathbf{w} \in H$. Conditioning on this event, $\ell_{\text{0-1}}(\mathbf{w}) > \eta + \epsilon$ yields a contradiction:

$$
\ell_{\text{0-1}}(\mathbf{w}) - \frac{\epsilon}{4} \leq \Pr_{(\mathbf{x},y)\sim_{\text{unif.}} S}[h_\mathbf{w}(\mathbf{x}) \neq y] \leq \Pr_{(\mathbf{x},y)\sim_{\text{unif.}} S}[h_{\hat{\mathbf{w}}}(\mathbf{x}) \neq y] \leq \ell_{\text{0-1}}(\hat{\mathbf{w}}) + \frac{\epsilon}{4} \leq \eta + \frac{3\epsilon}{4}. \quad\square
$$

We are now ready to state and prove our main theorem.

**Theorem 3** (Learning $\gamma$-margin Massart halfspaces). *Let $D$ be an instance of the $\eta$-Massart halfspace model with margin $\gamma$, and let $\epsilon, \delta \in (0, 1)$. Algorithm 1 with $T_1 \geq \frac{16}{\epsilon^2\gamma^2}\lceil\log_2(\frac{2}{\delta})\rceil$, $T_2 \geq \frac{8}{\epsilon^2}\log(\frac{4|T_1|}{\delta})$ returns $\mathbf{w}$ such that $\ell_{\text{0-1}}(\mathbf{w}) \leq \eta + \epsilon$ with probability $\geq 1 - \delta$, using $O((\epsilon\gamma)^{-2}\log(\delta^{-1}) + \epsilon^{-2}\log((\epsilon\gamma\delta)^{-1}))$ samples and $O(d\epsilon^{-4}\gamma^{-2}\log(\delta^{-1})\log((\epsilon\gamma\delta)^{-1}))$ time.*

*Proof.* First, applying Lemma 3 to each of the $N$ independent runs of Line 6 to Line 9 shows that the premise of Lemma 4 is met except with probability $\frac{\delta}{2}$. The correctness claim then follows from Lemma 4. The sample complexity is immediate, and the runtime bound follows because the bottleneck operation is computing the value of $h_\mathbf{w}(\mathbf{x})$ for all $(\mathbf{x}, y) \in S$ and $\mathbf{w} \in H$. $\qquad\square$

## 4 Massart generalized linear models

In this section, we present a key piece of intuition motivating our extension to the Massart GLM noise model (see Definition 2), deferring a full proof to Appendix A. In this setting, instead of being upper bounded by a fixed constant $\eta$, the noise rate is data-dependent and upper bounded by $\frac{1 - \sigma(\mathbf{w}^\star \cdot \mathbf{x})}{2}$ where $\sigma$ is odd non-decreasing and $\mathbf{w}^\star$ is the optimal halfspace. Inspired by our approach in Section 3, we propose a novel separating hyperplane based on the previously described reweighting scheme. We argue in this section that $\mathbf{E}_{\mathbf{x},y}\left[\frac{(\sigma(\mathbf{w}\cdot\mathbf{x}) - y)}{|\mathbf{w}\cdot\mathbf{x}|}\mathbf{x}\right]$ is a valid separating hyperplane, generalizing Lemma 1.

**Lemma 5** (Separating hyperplane for Massart GLMs). *Let $D$ be an instance of the $\sigma$-Massart GLM model with margin $\gamma$ (with respect to halfspace $\mathbf{w}^\star$), and $\mathbf{w} \in \mathbb{R}^d$ has $\ell_{\text{0-1}}(\mathbf{w}) \geq \mathrm{opt}_{\text{RCN}} + \epsilon$. We have that $\mathbf{E}_{(\mathbf{x},y)\sim D}\left[\frac{(\sigma(\mathbf{w}\cdot\mathbf{x}) - y)}{|\mathbf{w}\cdot\mathbf{x}|}\mathbf{x}\right] \cdot (\mathbf{w} - \mathbf{w}^*) \geq 2\epsilon$.*

*Proof Sketch.* We use the definition of sets $A, \beta(\mathbf{x})$ from Lemma 2. By expanding out the expression for $\ell_{0\text{-}1}(\mathbf{w})$ similarly to Lemma 2, we obtain that $\frac{1}{2} \cdot \mathbf{E}_{(\mathbf{x},y) \sim D} \left[ (|\sigma(\mathbf{w}^* \cdot \mathbf{x})| - \beta(\mathbf{x})) \mathbb{1}\{\mathbf{x} \in A\} \right] + \frac{1}{2} \cdot \mathbf{E}_{(\mathbf{x},y) \sim D} \left[ (|\sigma(\mathbf{w}^* \cdot \mathbf{x})| + \beta(\mathbf{x})) \mathbb{1}\{\mathbf{x} \notin A\} \right] \geq \epsilon$. Let $g(\mathbf{x}) = \frac{(\sigma(\mathbf{w} \cdot \mathbf{x}) - y)}{|\mathbf{w} \cdot \mathbf{x}|} \cdot (\mathbf{w} - \mathbf{w}^*)$. Now, we argue that $\mathbf{E}_{\mathbf{x} \sim D_\mathbf{x}}[g(\mathbf{x})]$ is greater than the left hand side of the previous inequality. We do a case analysis. For $\mathbf{x} \in A$, we observe that $g(\mathbf{x}) \geq (|\sigma(\mathbf{w}^* \cdot \mathbf{x})| - \beta(\mathbf{x})) \frac{|\mathbf{w} \cdot \mathbf{x}| - |\mathbf{w}^* \cdot \mathbf{x}|}{|\mathbf{w} \cdot \mathbf{x}|} \geq (|\sigma(\mathbf{w}^* \cdot \mathbf{x})| - \beta(\mathbf{x}))$. Here, we obtained the first inequality by adding and subtracting the corresponding term with $\sigma(\mathbf{w}^* \cdot \mathbf{x})$ and then using monotonicity. The final inequality follows from the fact that $\beta(\mathbf{x}) \geq |\sigma(\mathbf{w}^* \cdot \mathbf{x})|$. For $\mathbf{x} \notin A$, we obtain that $g(\mathbf{x}) = (|\sigma(\mathbf{w} \cdot \mathbf{x})| + \beta(\mathbf{x})) \frac{|\mathbf{w} \cdot \mathbf{x}| + |\mathbf{w}^* \cdot \mathbf{x}|}{|\mathbf{w} \cdot \mathbf{x}|} \geq (|\sigma(\mathbf{w}^* \cdot \mathbf{x})| + \beta(\mathbf{x}))$ where we obtain the inequality by doing a case analysis: (1) $|\mathbf{w} \cdot \mathbf{x}| \leq |\mathbf{w}^* \cdot \mathbf{x}|$, in this case $\frac{|\mathbf{w} \cdot \mathbf{x}| + |\mathbf{w}^* \cdot \mathbf{x}|}{|\mathbf{w} \cdot \mathbf{x}|} \geq 2$ and $2\beta(\mathbf{x}) \geq (|\sigma(\mathbf{w}^* \cdot \mathbf{x})| + \beta(\mathbf{x}))$ and (2), $|\mathbf{w} \cdot \mathbf{x}| \geq |\mathbf{w}^* \cdot \mathbf{x}|$, in this case $|\sigma(\mathbf{w} \cdot \mathbf{x})| \geq |\sigma(\mathbf{w}^* \cdot \mathbf{x})|$ and hence we are done. Now, taking the expectation of $g(\mathbf{x})$ completes the proof of the claim. $\qquad\square$

However, we are met with the same obstacle as before: $|\mathbf{w} \cdot \mathbf{x}|$ can be arbitrarily small. The previous approach of adding $\gamma$ to the denominator does not work immediately. Instead, we add the rescaled term $\frac{\epsilon}{2-\epsilon} \cdot \gamma$. Adding a smaller term in the denominator increases the bound on the norm of the increments in each step, resulting to a larger bound on the number of iterations (and, therefore, sample complexity) by a factor of $\epsilon^{-2}$. However, this rescaling is useful to obtain an analogue of Lemma 2 for the case of Massart GLMs (Lemma 6). The rescaling essentially accounts for the part of the distribution where $|\mathbf{w} \cdot \mathbf{x}|$ is smaller than $|\mathbf{w}^* \cdot \mathbf{x}|$ and the signs disagree. This is important because the size of $|\mathbf{w} \cdot \mathbf{x}|$ is quantitatively more significant in the Massart GLM case.

Combining this with a modified version of the Perspectron algorithm and analysis (see Algorithm 2), we obtain our final result with sample complexity $\widetilde{O}(\gamma^{-2}\epsilon^{-4})$ (see Theorem 4).

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

# A  Massart GLMs

In this section, we prove our result regarding learning in the Massart generalized linear model with a margin. Our analysis is similar to that of Section 3 but requires a modified version of Lemma 2, which gives a "bounded" separating hyperplane in the case of Massart GLM (Lemma 6). We first state the definition of our model again (slightly generalized to relax the assumption that $\sigma$ is odd).

**Definition 3** (Massart GLM). *Let $\sigma : [-1, 1] \to [-1, 1]$ be a non-decreasing function. We say that a distribution $D$ on $\mathbb{B}^d \times \{\pm 1\}$ is an instance of the $\sigma$-Massart generalized linear model (GLM) with constant shift $\tau$ and margin $\gamma$ with respect to $\mathbf{w}^\star$ if the following conditions hold.*

- $\big| |\sigma(t)| - |\sigma(-t)| \big| \le \tau$ *for all* $t \in [0, 1]$.

- *There exists* $\mathbf{w}^\star \in \mathbb{R}^d$ *such that* $\|\mathbf{w}^\star\| = 1$ *and* $\mathbf{Pr}[|\mathbf{w}^\star \cdot \mathbf{x}| < \gamma] = 0$.

- *For all* $\mathbf{x} \in \mathrm{supp}(D_{\mathbf{x}})$*, there is an* $\eta(\mathbf{x}) \in [0, \frac{1 - |\sigma(\mathbf{w}^\star \cdot \mathbf{x})|}{2}]$ *such that*

$$\Pr_{y \sim D_y(\mathbf{x})} [y \neq h_{\mathbf{w}^\star}(\mathbf{x})] = \eta(\mathbf{x}).$$

We state and prove the following lemma which provides a separating hyperplane in this setting.

**Lemma 6** (Separating Hyperplane). *Let $D$ be a distribution on $\mathbb{R}^d \times \{\pm 1\}$ such satisfying Definition 3. Let $\mathbf{w} \in \mathbb{R}^d$ be such that $\mathbf{Pr}_{(\mathbf{x}, y) \sim D}[\mathrm{sign}(\mathbf{w} \cdot \mathbf{x}) \neq y] \ge \mathbf{E}_{\mathbf{x} \sim D_{\mathbf{x}}} \big[ \frac{1 - |\sigma(\mathbf{w}^* \cdot x)| + \tau}{2} \big] + \epsilon$. Then, we have*

$$\mathbf{E}_{(\mathbf{x}, y) \sim D} \left[ (\sigma(\mathbf{w}.\mathbf{x}) - y) \frac{\mathbf{x}}{|\mathbf{w} \cdot \mathbf{x}| + \alpha\gamma} \right] \cdot (\mathbf{w} - \mathbf{w}^*) \ge \epsilon \,, \ \textit{for } \alpha = \epsilon/(2 - \epsilon).$$

*Proof.* Let $A = \{\mathbf{x} \in \mathbb{R}^d \mid \mathrm{sign}(\mathbf{w} \cdot \mathbf{x}) = \mathrm{sign}(\mathbf{w}^* \cdot \mathbf{x})\}$ and $B = \{\mathbf{x} \in \mathbb{R}^d \mid \mathrm{sign}(\mathbf{w} \cdot \mathbf{x} \neq \mathrm{sign}(\mathbf{w}^* \cdot \mathbf{x})\}$. We have that

$$
\begin{aligned}
I &= \mathbf{E}_{(\mathbf{x}, y) \sim D} \left[ (\sigma(\mathbf{w} \cdot \mathbf{x}) - y) \frac{\mathbf{x}}{|\mathbf{w} \cdot \mathbf{x}| + \alpha\gamma} \right] \cdot (\mathbf{w} - \mathbf{w}^*) \\
&= \mathbf{E}_{\mathbf{x} \sim D_{\mathbf{x}}} \left[ (\sigma(\mathbf{w} \cdot \mathbf{x}) - \beta(\mathbf{x})\mathrm{sign}(\mathbf{w}^* \cdot \mathbf{x})) \cdot \frac{(\mathbf{w} \cdot \mathbf{x} - \mathbf{w}^* \cdot \mathbf{x})}{|\mathbf{w} \cdot \mathbf{x}| + \alpha\gamma} \right]
\end{aligned}
$$

Define $g(\mathbf{x}) = (\sigma(\mathbf{w} \cdot \mathbf{x}) - \beta(\mathbf{x})\mathrm{sign}(\mathbf{w}^* \cdot \mathbf{x})) \cdot \frac{(\mathbf{w} \cdot \mathbf{x} - \mathbf{w}^* \cdot \mathbf{x})}{|\mathbf{w} \cdot \mathbf{x}| + \alpha\gamma}$. We analyze $g(\mathbf{x})$ separately for $\mathbf{x} \in A$ and $\mathbf{x} \in B$.

First consider points $\mathbf{x}$ in $A$. For any $\mathbf{x} \in A$, we have that

$$
\begin{aligned}
g(\mathbf{x}) &= \big( \sigma(\mathbf{w} \cdot \mathbf{x}) - \sigma(\mathbf{w}^* \cdot \mathbf{x}) + \sigma(\mathbf{w}^* \cdot \mathbf{x}) - \beta(\mathbf{x})\mathrm{sign}(\mathbf{w}^* \cdot \mathbf{x}) \big) \cdot \frac{(\mathbf{w} \cdot \mathbf{x} - \mathbf{w}^* \cdot \mathbf{x})}{|\mathbf{w} \cdot \mathbf{x}| + \alpha\gamma} \\
&\ge \big( |\sigma(\mathbf{w}^* \cdot \mathbf{x})|\mathrm{sign}(\mathbf{w}^* \cdot \mathbf{x}) - \beta(\mathbf{x})\mathrm{sign}(\mathbf{w}^* \cdot \mathbf{x}) \big) \cdot \frac{(\mathbf{w} \cdot \mathbf{x} - \mathbf{w}^* \cdot \mathbf{x})}{|\mathbf{w} \cdot \mathbf{x}| + \alpha\gamma} \\
&\ge \big( |\sigma(\mathbf{w}^* \cdot \mathbf{x})| - \beta(\mathbf{x}) \big) \cdot \frac{|\mathbf{w} \cdot \mathbf{x}| - |\mathbf{w}^* \cdot \mathbf{x}|}{|\mathbf{w} \cdot \mathbf{x}| + \alpha\gamma} \ge |\sigma(\mathbf{w}^* \cdot \mathbf{x})| - \beta(\mathbf{x}) \,.
\end{aligned}
$$

The second inequality follows from the fact that $(\sigma(\mathbf{w} \cdot \mathbf{x}) - \sigma(\mathbf{w}^* \cdot \mathbf{x})) \cdot (\mathbf{w} \cdot \mathbf{x} - \mathbf{w}^* \cdot \mathbf{x}) \ge 0$ since $\sigma$ is monotonically increasing. The third inequality holds because $\mathrm{sign}(\mathbf{w} \cdot \mathbf{x}) = \mathrm{sign}(\mathbf{w}^* \cdot \mathbf{x})$. The final inequality is true because $\frac{|\mathbf{w} \cdot \mathbf{x}| - |\mathbf{w}^* \cdot \mathbf{x}|}{|\mathbf{w} \cdot \mathbf{x}| + \alpha\gamma} \le 1$ and $\beta(\mathbf{x}) \ge |\sigma(\mathbf{w}^* \cdot \mathbf{x})|$.

We now consider the case where $\mathbf{x} \in B$. Since $\mathrm{sign}(\mathbf{w} \cdot \mathbf{x}) \neq \mathrm{sign}(\mathbf{w}^* \cdot \mathbf{x})$, we have that $g(\mathbf{x}) = \big( |\sigma(\mathbf{w} \cdot \mathbf{x})| + \beta(\mathbf{x}) \big) \cdot \frac{|\mathbf{w} \cdot \mathbf{x}| + |\mathbf{w}^* \cdot \mathbf{x}|}{|\mathbf{w} \cdot \mathbf{x}| + \alpha\gamma}$. We split $B$ into two finer regions. Define $B_1 = \{\mathbf{x} \in B \mid |\mathbf{w} \cdot \mathbf{x}| \ge |\mathbf{w}^* \cdot \mathbf{x}|\}$ and $B_2 = \{\mathbf{x} \in B \mid |\mathbf{w} \cdot \mathbf{x}| < |\mathbf{w}^* \cdot \mathbf{x}|\}$. First, consider $\mathbf{x} \in B_1$. We have that $|\sigma(\mathbf{w} \cdot \mathbf{x})| \ge \max(0, |\sigma(-\mathbf{w} \cdot \mathbf{x})| - \tau)$. We also have that $|\sigma(-\mathbf{w} \cdot \mathbf{x})| \ge |\sigma(\mathbf{w}^* \cdot \mathbf{x})|$ since $|\mathbf{w} \cdot \mathbf{x}| \ge |\mathbf{w}^* \cdot \mathbf{x}|$ and $\sigma$ is monotone non-decreasing. Also, observe that $\frac{|\mathbf{w} \cdot \mathbf{x}| + |\mathbf{w}^* \cdot \mathbf{x}|}{|\mathbf{w} \cdot \mathbf{x}| + \alpha\gamma} \ge 1$ since $|\mathbf{w}^* \cdot \mathbf{x}| \ge \gamma$. Thus, we obtain that $g(\mathbf{x}) \ge \max(0, |\sigma(\mathbf{w}^* \cdot \mathbf{x})| - \tau) + \beta(\mathbf{x})$. Finally, we consider

$\mathbf{x} \in B_2$. Let $c(\mathbf{x}) = \frac{|\mathbf{w}^* \cdot \mathbf{x}|}{|\mathbf{w} \cdot \mathbf{x}|}$. We have that

$$g(\mathbf{x}) \geq \beta(\mathbf{x}) \frac{|\mathbf{w}^* \cdot \mathbf{x}| + |\mathbf{w} \cdot \mathbf{x}|}{|\mathbf{w} \cdot \mathbf{x}| + \alpha\gamma} \geq \beta(\mathbf{x}) \frac{1 + \frac{|\mathbf{w}^* \cdot \mathbf{x}|}{|\mathbf{w} \cdot \mathbf{x}|}}{1 + \alpha \frac{\gamma}{|\mathbf{w}^* \cdot \mathbf{x}|} \frac{|\mathbf{w}^* \cdot \mathbf{x}|}{|\mathbf{w} \cdot \mathbf{x}|}} \geq \beta(\mathbf{x}) \frac{1 + c(\mathbf{x})}{1 + \alpha c(\mathbf{x})} \geq (2 - \epsilon)\beta(\mathbf{x}).$$

The third inequality follows from the fact that $|\mathbf{w}^* \cdot \mathbf{x}| \geq \gamma$ and the last inequality is true because $\frac{1+c}{1+\alpha c} \geq 2 - \epsilon$ for any $c \geq 1$ when $\alpha = \frac{\epsilon}{2-\epsilon}$. Since $1 \geq \beta(\mathbf{x}) \geq |\sigma(\mathbf{w}^* \cdot x)|$, we have that $g(\mathbf{x}) \geq \beta(\mathbf{x}) + |\sigma(\mathbf{w}^* \cdot \mathbf{x})| - \epsilon$.

Thus, we obtain that

$$I \geq \mathop{\mathbf{E}}_{(\mathbf{x},y)\sim D}[(|\sigma(\mathbf{w}^* \cdot \mathbf{x})| - \beta(\mathbf{x}))\mathbb{1}_{\{\mathbf{x}\in A\}}]$$
$$+ \mathop{\mathbf{E}}_{(\mathbf{x},y)\sim D}[(\max(0, |\sigma(\mathbf{w}^* \cdot \mathbf{x})| - \tau) + \beta(\mathbf{x}))\mathbb{1}_{\{\mathbf{x}\in B\}}] - \epsilon \qquad (7)$$

We now use our assumption on the error of $\mathbf{w}$. We have that

$$\epsilon \leq \mathbf{Pr}_{(\mathbf{x},y)\sim D}[\mathrm{sign}(\mathbf{w} \cdot \mathbf{x}) \neq y] - \mathop{\mathbf{E}}_{\mathbf{x}\sim D_\mathbf{x}}\left[\frac{1 - |\sigma(\mathbf{w}^* \cdot \mathbf{x})|}{2}\right]$$

$$= \mathop{\mathbf{E}}_{(\mathbf{x},y)\sim D}\left[\mathbb{1}\{\mathbf{x} \in A\}\frac{1 - \beta(\mathbf{x})}{2}\right] + \mathop{\mathbf{E}}_{(\mathbf{x},y)\sim D}\left[\mathbb{1}\{\mathbf{x} \in B\}\frac{1 + \beta(\mathbf{x})}{2}\right] - \mathop{\mathbf{E}}_{\mathbf{x}\sim D_\mathbf{x}}\left[\frac{1 - |\sigma(\mathbf{w}^* \cdot \mathbf{x})|}{2}\right]$$

$$= \frac{1}{2} \cdot \mathop{\mathbf{E}}_{(\mathbf{x},y)\sim D}\left[(|\sigma(\mathbf{w}^* \cdot \mathbf{x})| - \beta(\mathbf{x}))\mathbb{1}\{\mathbf{x} \in A\}\right]$$

$$+ \frac{1}{2} \cdot \mathop{\mathbf{E}}_{(\mathbf{x},y)\sim D}\left[(\max(0, |\sigma(\mathbf{w}^* \cdot \mathbf{x})| - \tau)\beta(\mathbf{x}))\mathbb{1}\{\mathbf{x} \in B\}\right]$$

Plugging this into Equation (7), we obtain that $I \geq 2\epsilon - \epsilon \geq \epsilon$. This completes the proof. $\qquad \square$

We can now prove our main theorem about Massart GLMs. The algorithm we use, Algorithm 2, is a modified version of the Perspectron algorithm (Algorithm 1), where we subsitute the value of the parameter $\gamma$ with $\gamma \cdot \frac{\epsilon}{2-\epsilon}$ and the updates involve the function $\sigma$.

**Theorem 4.** *Let $D$ be an instance of the $\sigma$-Massart GLM with constant shift $\tau$ and margin $\gamma$ with respect to $\mathbf{w}^\star$, and let $\epsilon, \delta \in (0, 1)$. Algorithm 2 with , $T_1 \geq \left(\frac{32}{\epsilon^4 \gamma^2}\right)\lceil\log_2(\frac{2}{\delta})\rceil$, $T_2 \geq \frac{8}{\epsilon^2}\log(\frac{4|T_1|}{\delta})$ returns $\mathbf{w}$ such that $\ell_{0\text{-}1}(\mathbf{w}) \leq \mathbf{E}_{\mathbf{x}\sim D_\mathbf{x}}\left[\frac{1 - |\sigma(\mathbf{w}^* \cdot \mathbf{x}| + \tau}{2}\right] + \epsilon$ with probability $\geq 1 - \delta$, using*

$$O\left(\frac{\log(\frac{1}{\delta})}{\epsilon^4 \gamma^2} + \frac{\log(\frac{1}{\epsilon\gamma\delta})}{\epsilon^2}\right) \text{ samples and } O\left(\frac{d\log(\frac{1}{\delta})\log(\frac{1}{\epsilon\delta})}{\epsilon^6 \gamma^2}\right) \text{ time.}$$

---

**Algorithm 2:** GLMPerspectron

---
**1 Input:** $\{\mathbf{x}^i, y^i\}_{i\in[T_1+T_2]} \subset \mathbb{R}^d \times \{\pm 1\}$ drawn i.i.d. from $D$ in the $\sigma$-Massart GLM model with margin $\gamma$, parameter $\alpha \in (0, 1)$, step size $\lambda > 0$, failure probability $\delta \in (0, \frac{1}{2})$

**2** $\beta \leftarrow 1 - 2\eta$, $N \leftarrow \lceil\log_2(\frac{2}{\delta})\rceil$, $T \leftarrow \lceil\frac{T_1}{N}\rceil$

**3** $H \leftarrow \emptyset$

**4 for** $j \in [N]$ **do**

**5** $\quad$ $\mathbf{w}^{1,j} \leftarrow \mathbb{0}_d$

**6** $\quad$ **for** $t \in [\min(T, T_1 - (j-1)T)]$ **do**

**7** $\quad\quad$ $i \leftarrow (j-1)T + t$

**8** $\quad\quad$ $\mathbf{w}^{t+1,j} \leftarrow \mathbf{w}^{t,j} - \lambda\frac{\sigma(\mathbf{w}^{t,j}\cdot\mathbf{x}^i) - y^i}{|\mathbf{w}^{t,j}\cdot\mathbf{x}^i| + \gamma\cdot\alpha}\mathbf{x}^i$

**9** $\quad$ **end**

**10** $\quad$ $H \leftarrow H \cup \{\mathbf{w}^{t,j}\}_{t\in[\min(T, T_1-(j-1)T)]}$

**11 end**

**12** $S \leftarrow \{\mathbf{x}^i, y^i\}_{i=T_1+1}^{T_1+T_2}$

**13** $\mathbf{w} \leftarrow \arg\min_{\mathbf{w}\in H} \mathbf{Pr}_{(\mathbf{x},y)\sim\text{unif.} S}[h_\mathbf{w}(\mathbf{x}) \neq y]$

**14 Return:** $h_\mathbf{w}$

---

*Proof.* Given Lemma 6, the first step of the proof is exactly analogous to the proof of Lemma 3, i.e., we can show the following claim.

**Claim 1.** *Let $\{\mathbf{x}^i, y^i\}_{i \in [T]} \sim_{i.i.d.} D$, where $D$ satisfies Definition 3. Consider iterating, from $\mathbf{w}^1 := \mathbb{0}_d$,*

$$\mathbf{w}^{t+1} \leftarrow \mathbf{w}^t - \lambda \frac{\sigma(\mathbf{w}^t \cdot \mathbf{x}^t) - y^t}{|\mathbf{w}^t \cdot \mathbf{x}^t| + \gamma\epsilon/(2-\epsilon)} \mathbf{x}^t, \tag{8}$$

*for $\lambda := \frac{\gamma\epsilon}{(2-\epsilon)\sqrt{2T}}$. Then if $T \geq \frac{32}{\epsilon^4\gamma^2}$, $\mathbf{Pr}[\min_{t \in [T]} \ell_{0\text{-}1}(\mathbf{w}^t) \geq \mathbf{E}_{\mathbf{x} \sim D_{\mathbf{x}}}\left[\frac{1-|\sigma(\mathbf{w}^* \cdot \mathbf{x}| + \tau}{2}\right] + \frac{\epsilon}{2}] \leq \frac{1}{2}$.*

The proof is analogous to the proof of Lemma 3, but we use $\gamma\epsilon/(2-\epsilon)$ in the place of $\gamma$. To amplify the success probability and finish the proof, we once more use Lemma 4. □

# B Omitted proofs

## B.1 Learning with unknown noise rate

In this section, we can obtain the same sample complexity (upto logarithmic factors) as Theorem 3 even when the noise rate $\eta$ is unknown to the learner. The argument is the following: we argue that our separating hyperplane (Lemma 2) is tolerant to $O(\epsilon)$ noise in the parameter $\eta$. We can then discretize the interval $[0, 1/2]$ into intervals of size $\epsilon$ and run the training algorithm multiple times for these different choices. Then, we can output the hypothesis with lowest validation error among the classifiers output by these different runs of the algorithm. We now argue that this idea indeed works.

**Lemma 7.** *If $D$ is an instance of the $\eta$-Massart halfspace model with margin $\gamma$ with respect to $\mathbf{w}^\star$, and $\mathbf{w} \in \mathbb{R}^d$ has $\ell_{0\text{-}1}(\mathbf{w}) \geq \eta + \epsilon$,*

$$\mathbf{E}_{(\mathbf{x},y)\sim D}\left[(\tilde{\beta}\mathrm{sign}(\mathbf{w} \cdot \mathbf{x}) - y)\frac{\mathbf{x}}{|\mathbf{w} \cdot \mathbf{x}| + \gamma}\right] \cdot (\mathbf{w} - \mathbf{w}^*) \geq 2\epsilon, \text{ for } \tilde{\beta} \in ((1 - 2\eta) - \epsilon, (1 - 2\eta)].$$

*Proof.* The proof is almost identical to the proof of Lemma 2 except for a few steps. We highlight the differences. We reuse the notation from the previous proof.

First consider $\mathbf{x} \notin A$, we observe that using the same argument as before, we now obtain

$$g(\mathbf{x}) \geq \tilde{\beta} + \beta(\mathbf{x}) \geq \beta + \beta(\mathbf{x}) - \epsilon.$$

In the case of $\mathbf{x} \in A$, since $\beta(\mathbf{x}) \geq \beta \geq \tilde{\beta}$, we obtain

$$g(\mathbf{x}) \geq (\tilde{\beta} - \beta(\mathbf{x})) \geq \beta - \beta(\mathbf{x}) - \epsilon.$$

Using this, we can can complete the proof by repeating the steps of the previous proof. □

Having proved this, our algorithm is simple, run over the $(1/(2\epsilon))$ choices of $\tilde{\beta}$ in $[0, 1/2]$ and run the algorithm from Theorem 3 for each choice, reusing the same samples in the different run's of the algorithm. The correctness follows Lemma 7 and the proof of Theorem 3 since one of the the choices of parameters must lie in the interval $((1 - 2\eta) - \epsilon, (1 - 2\eta)]$.

