# OpenReview forum: "Learning Noisy Halfspaces with a Margin: Massart is No Harder than Random"
_NeurIPS.cc/2024/Conference — NeurIPS 2024 spotlight_

### Official Review · Reviewer_x3Ci · 2024-06-29

**Soundness:** 3
**Presentation:** 3
**Contribution:** 3
**Rating:** 7
**Confidence:** 2

**Summary:**

The paper studies learning half-spaces in the Massart noise setting. That is for $\gamma\eta >0$ and $\eta(x)\leq \eta $ for all $x\in supp(D_{x})$,  half space $w^*$ and such that $P( | w^* x  | \geq \gamma)$ and $P ( sign( w^* x ) \not= y)=\eta(x)$ for all $x\in supp(D_{x})$ - given such an instance and $\varepsilon>0$ the paper now states the problem of finding a half spaces $w$ such that $P[sign(wx)\not=y]\leq \eta+\varepsilon]$ using as few samples and computations as possible. The paper achieves a new bound on the number of samples needed to solve this problem of $\tilde{O}(\gamma\varepsilon)^{-2}$ (suppressing factors of  $\log(1/\delta)$). This is an improvement over previous results which uses $\gamma^{-3}\varepsilon^{-5}$, $\gamma^{-4}\varepsilon^{-3}$. They also show it in a more general setting for $\sigma$ odd and non decreasing where $P[sign(w*x)\not= y]=\frac{1-\sigma(wx)}{2}$ for all $x\in supp(D_{x})$. They obtain a sample complexity bound of $(\tilde{O})((\gamma\varepsilon)^{-2})$ here.

**Strengths:**

Originality:
The paper seems to combine many ideas from different papers and in this way get a novel result and improvement over previous results. Related work is well cited.

Quality:
The main of the paper includes a full proof of the Massart noise and a proof sketch of the general setting. The proof of the Massart noise case on a first quick read seems ok. The proof sketch is ok - I have not read the appendix with the full proof so I will not comment on the technically soundness of the proof. The authors also address the limits of their work and state interesting future work.

Clarity:
The paper is very well written and explains very well the ideas behind the proof.

Significance:
The improvement in sample complexity is a polynomial improvement in $(\gamma\varepsilon)^{-1}$ so I would say significant.

**Weaknesses:**

Weaknesses:
Not anything significant - related to the question below do the other papers also use $\eta$ in the error bound instead of $E[\eta(x)]\leq \eta$? If yes why not use $E[\eta(x)]$?

**Questions:**

\section*{Questions}

Line 18-20: I really like how it was $Pr_{x\sim D_{x}}$ was used as it made the randomness explicit - I didn't at first get that the randomness in the $Pr$ (line 20) was over $y\sim D_{y}(x)$ I don't know if other readers would have the same experience.

Line 72 : what error does previous work in massart noise setting use? If other- why not use the same?


Line 102 $\eta(x)$?

Not consistent whether it is $Pr_{\cdot}$ or just $Pr$

Table: Good table. Is the running time not interesting?

The real numbers $R$ and natural numbers $N$ didn't look like I'm used to seeing them - I don't know if this is on purpose if it is don't mind changing it - I just wanted to let you know in case that something has complied weirdly.

Line 221: should it be $B^d\times \{-1,1\}$?

Line 229: why isn't it a vector? Can you give the derivation of this expression?

Line 238: $h_{w'}=sign((w..$ shouldnt it be $h_{w'}=sign((w'..$

Line 261: what does the $l_\eta(w*,x,h)$ mean? Happens more places in this proof.

Line 263: Please show fact (1), or should it be $l_\eta(-yw*x)$. If facts 1) and 2) follow from the fact stated at the beginning of the proof from [DGT19b] maybe consider moving these together.

Line 276: it says $w^{\star}$ instead of $w^*$

Line 298-299: The third line in the equation missing $x^t$

Line 289 algorithm line 7: this makes $w^t$ and $x^t$ independent right? Which is used in the proof of lemma 3  last inequality of the large equation combined with lemma 2?

Line 307: why say anything about contradiction - doesn't it just follow from the inequality afterward?

Line 313-316: I guess It doesn't matter that the last badge isn't necessarily of size $T$? Or is it necessarily of size $T$?

Line 329: what is $B$?

Would it be interesting to make the computational cost independent of d with the cost of polynomial factors in $1/\gamma\varepsilon$?

**Limitations:**

They address the limitations of the work.

---

> ### Author Rebuttal · Authors · 2024-08-05
>
> Thank you for your careful reading, and all of your detailed feedback. We are glad that you found our explanations clear. Regarding improved dependences on $\mathbb{E}[\eta(\mathbf{x})]$, prior work has shown that such guarantees are likely computationally intractable for statistical query based algorithms (such as ours, and all previous Massart learners) under standard complexity assumptions: see, for example, the citations [CKMY20, DK22, DKMR22, NT22]. Therefore, to give a polynomial-time algorithm, we focus on achieving error $\eta + \epsilon$. We discuss this point in Lines 73-76 of our submission. This is consistent with previous works in the Massart model, re: your question about Line 72.
>
> We now address the reviewer's other more specific questions.
>
> Line 102: This should be $\eta$, as stated, as it is about the noise rate upper bound.
> Table: All runtimes are nearly-linear in the input size, but we will add such a remark.
>
> Line 229: This was a typo, and there should be an extra factor of $\mathbf{x}$ due to chain rule, so it is a vector. Thank you for noticing this.
>
> Line 261: This was a typo, it should say $\ell_\eta(-y\mathbf{w}^\star \cdot \mathbf{x})$, and we will fix this.
>
> Line 263: Yes, as you point out, the expression should be about $\mathbf{w}^\star$.
>
> Line 289: This is correct, and we will note this for clarity.
>
> Line 307: Good catch, we will make this simplification.
>
> Line 313-316: In our application of Algorithm 1 (Theorem 3), we ensured even divisibility.
>
> Line 329: $B$ was intended to be the complement of $A$ (as in Lemma 2); we will clarify this.
>
> Computational cost: We expect that achieving independence of $d$ is unachievable, as one needs to at least read in the samples. However, we do agree that it is interesting to see if fast dimensionality reduction techniques could be used to speed up the low-order terms of our algorithm's runtime; thank you for this suggestion.
>
> *[CKMY20] Sitan Chen, Frederic Koehler, Ankur Moitra and Morris Yau. Classification Under Misspecification: Halfspaces, Generalized Linear Models, and Connections to Evolvability. NeurIPS 2020*
>
> *[DK22] Ilias Diakonikolas, Daniel Kane. Near-Optimal Statistical Query Hardness of Learning Halfspaces with Massart Noise. Conference on Learning Theory, 2022*
>
> *[DKMR22] Ilias Diakonikolas, Daniel Kane, Pasin Manurangsi, and Lisheng Ren, Cryptographic Hardness of Learning Halfspaces with Massart Noise. NeurIPS 2022*
>
> *[NT22] Rajai Nasser, Stefan Tiegel. Optimal SQ Lower Bounds for Learning Halfspaces with Massart Noise. Conference on Learning Theory, 2022*

---

> ### Comment · Reviewer_x3Ci · 2024-08-08
>
> Thanks for your response. My understanding of the change due to the technical issue in terms of Table 1: the Massart noise column is now two columns: Massart with sample complexity $(\gamma\varepsilon)^{-2}$ and Massart GLM with sample complexity $\gamma^{-2}\varepsilon^{-4}$ for your row. If this is correct please “fill in” the sample complexity that [DGT19b] and [CKMY20] get for Massart and Massart GLM (or other relevant references). To this end please remind me if $\varepsilon$ is always less than $\gamma$.

---

> > ### Author Response · Authors · 2024-08-09
> > **Reply to reviewer x3Ci**
> >
> > Table 1 in our paper is currently only about Massart Halfspaces. We are happy to add one more column with the sample complexity of Massart GLM. This column will have the sample complexity of $\gamma^{-2}\epsilon^{-4}$ in our row, as the reviewer notes.
> >
> > ### In regards to the bounds obtained by prior work:
> > 1) [DGT19b] do not study this model.
> > 2) [CKMY20] consider this model with an additional restriction on $\sigma$ being $L$-Lipschitz and $|\sigma(\mathbf{w}^*\cdot\mathbf{x})|\geq \gamma$ for all $\mathbf{x}$. In this regime, they obtain a sample complexity of at least $L^4\gamma^{-4}\epsilon^{-6}$. In this regime, our result translates to a bound of $L^2\gamma^{-2}\epsilon^{-4}$ which is a strict improvement.
> >
> > ### Comparison between $\epsilon$ and $\gamma$:
> >
> > In general, these parameters are incomparable depending on situation (one is about a suboptimality guarantee, and one is a geometric assumption about the distribution). However, there are natural scenarios where one would prefer a worse $\epsilon$ dependence (as in our updated result) compared to a worse $\gamma$ dependence:
> >
> > 1) $\gamma$ is a high-dimensional parameter, so it is natural to have it depend on the dimension $d$. In particular, for natural models (e.g. $D$ is TV-close to the uniform distribution over the unit sphere) we really do have margin about $1/\sqrt{d}$.
> > 2)  $\epsilon$ measures error in the 0-1 output space, so it is natural to have it not depend on $d$ (i.e. treat it as a constant / free parameter).
> >
> > Concretely, in the margin-free setting, state-of-the-art algorithms all apply pre-processing techniques [BFKV96,DV04,DKT21] that transform an underlying distribution to one with margin effectively equal to $\Omega(\text{poly}(1/d))$. In these cases, a worse dependence on $1/\epsilon$ may be preferred over a bad dependence on $1/\gamma$.
> >
> > *[BFKV96] 	Avrim Blum, Alan M. Frieze, Ravi Kannan and Santosh S. Vempala. A Polynomial-Time Algorithm for Learning Noisy Linear Threshold Functions. FOCS 1996*
> >
> > *[DV04] John Dunagan and Santosh Vempala. A simple polynomial-time rescaling algorithm for solving linear programs. STOC 2004*
> >
> > *[DKT21] Ilias Diakonikolas, Daniel M. Kane and Christos Tzamos. Forster decomposition and learning halfspaces with noise. NeurIPS 2021*

---

> ### Comment · Reviewer_x3Ci · 2024-08-09
>
> Arrh good -  to this end, so [CKMY20] achieves sample complexity and runtime of $\Omega(L^4\gamma^{-4}\varepsilon^{-6})$ - I think line 113 only says something about runtime?(sorry if im mistaking).

---

> > ### Author Response · Authors · 2024-08-09
> >
> > Yes, that was a mistake in our description -- Line 113 should say the sample complexity of [CKMY20] is $\Omega(L^4 \gamma^{-4} \epsilon^{-6})$, not its runtime. This is directly comparable to our updated sample complexity of $\widetilde{O}(L^2 \gamma^{-2} \epsilon^{-4})$. Both runtimes pay an overhead in the dimension $d$ compared to the sample complexity, since they perform vector operations. Thank you for this catch.

---

> > > ### Comment · Reviewer_x3Ci · 2024-08-10
> > >
> > > Thanks for answering my questions and clearing up my misunderstandings - good luck!

---

### Official Review · Reviewer_vx3q · 2024-07-06

**Soundness:** 3
**Presentation:** 4
**Contribution:** 4
**Rating:** 7
**Confidence:** 4

**Summary:**

The paper considers the problem of PAC-learning $\gamma$-margin halfspaces under $\eta$-Massart noise. The paper provides an efficient algorithm achieving error $\eta+\epsilon$ with sample complexity $\tilde{O}(1/(\epsilon^2\gamma^2))$. The individual dependence  of the sample complexity on $\epsilon$ and $\gamma$ appears to be optimal for efficient algorithms in the light of lower bounds provided in previous work.

The algorithm is a form of stochastic-gradient descent where the loss function is the LeakyReLu loss. The gradient is carefully weighted with an appropriate weight in order to yield the desired results.

The paper also provides an algorithm for a more general model, the generalized linear model (GLM), under Massart-noise.

The paper is generally very-well written.

**Strengths:**

The problem of learning half-spaces under Massart noise is a fundamental problem that has received a lot of attention recently. This paper provides a simple algorithm with a low sample-complexity that is probably optimal.

The paper also provides generalizations to the GLM models.

**Weaknesses:**

I haven't found any significant weaknesses.

Typos:
- Page 3, line 97: “observe that that if” ->  “observe that if”

**Questions:**

- In Lemma 3, the parameters $\lambda$ and $T$ were chosen so that $\mathbb{P}[\mathcal{E}_T]\leq 1/2$, which necessitated the outer loop $j\in[N]=[\log_2(2/\delta)]$ to get success probability at least $1-\delta$. Wouldn't it be possible to choose the constant factors in $\lambda$ and $T$ differently to directly get  $\mathbb{P}[\mathcal{E}_T]\leq \delta$?

**Limitations:**

Since this is a theoretical paper, I do not see any potential negative societal impact.

---

> ### Author Rebuttal · Authors · 2024-08-05
>
> We appreciate that you found our paper well-written. Thank you for your typo suggestions; we will fix these in a revision. Re: your question in Lemma 3, we chose this parameter tradeoff because it yields a $\log(\frac 1 \delta)$ overhead in our runtime. As you suggest, it is also possible to directly obtain a failure probability of $\delta$ in one shot. However, this would require taking a number of steps polynomial in $\frac 1 \delta$, which is a worse overall tradeoff.

---

> > ### Comment · Reviewer_vx3q · 2024-08-11
> >
> > Thank you for the clarification. I remain my score.

---

### Official Review · Reviewer_kyaH · 2024-07-12

**Soundness:** 3
**Presentation:** 2
**Contribution:** 3
**Rating:** 7
**Confidence:** 3

**Summary:**

This submission studies the problem of learning halfspaces and generalized linear model in the Massart model under a margin assumption. In particular, for the case of halfspaces, the submission gives an efficient algorithm achieving (conjecturally optimal) error $\eta + \varepsilon$ using only $\tilde{O}(\gamma^{-2} \varepsilon^{-2})$ samples, where $\gamma$ is the margin parameter, and $\eta$ the upper bound on the noise rate. This matches what is known in the more benign RCN model. All previously known efficient algorithms in the Massart model require a number of samples that is polynomially worse in $\varepsilon$ and $\gamma$.

**Strengths:**

The Massart model is a semi-random noise model motivated by the question whether known (efficient) algorithmic approaches for the fully random model, in this case random classification noise (RCN), are overfitting to the specific aspects of the model. The work of [DGT19] designed the first efficient algorithm to non-trivially learn halfspaces in this harsher noise model (without margin assumption). The current work shows that under a margin assumption, this is possible with as few samples as known efficient algorithms for the more benign RCN model needs. All prior works were loose by polynomial factors in $\gamma$ and $\varepsilon$.

On a technical level the submission combines previous algorithmic approaches with a regret-minimization scheme leading to an elegant analysis and ultimately a better sample complexity.




[DGT19]: I. Diakonikolas, T. Gouleakis, and C. Tzamos. Distribution-independent PAC learning of halfspaces with massart noise

**Weaknesses:**

The presentation of the submission is generally solid but can be improved in several aspects. Below are comments aiming at this. Especially the comments related to the technical overview are important in order to understand why the submission is able to improve over previous work. I'm willing to increase my score to 6 or 7 if the authors agree to address the comments below and in particular, explain in their rebuttal how they would address the comments for the technical overview.


## Technical Overview

I like the structure of the technical overview and that it tries to highlight differences with previous works (this also shows expertise of the authors). Unfortunately however, in several places the writing is unclear (see below). I find it nice that the proof for the halfspace result is so short that it fits in the main body. Nevertheless, I would find a more extensive technical overview with commentary much more illustrative then including the full proof (the many formulas make this hard to read and understand what is going on).

Here are some specific comments about the technical overview:

- lines 129-136: It is not clear from the discussion why this approach incurs a too high sample complexity on a quantitative level.
- lines 153-162: It is not at all clear from the discussion why the proposed update rule improves the dependence on $\gamma$
- lines 172-174: You say that this approach also works for the case of massart halfsapces. Why did you not follow this approach? From the discussion it seems it would significantly simplify a part of the analysis. Is there anything else that breaks?


## Introduction

- lines 46-58: Before diving into the details of the fine-grained aspects and to set the stage, it would be helpful to briefly explicitly recall what is known in the general Massart model (without margin assumption) -- e.g., error $\eta$ is possible and this is likely optimal
- when you say "learn up to error $\varepsilon$" in the above paragraph, do you mean error $\eta + \varepsilon$?
- I believe the work [DKMR22] is not mentioned at all, this should be added
- Similarly, when talking about impossibility results in the agnostic model, the work [Tie23] (see also [DKMR22]) should be mentioned
- for some of the citations, the arxiv version is cited. Why not cite the conference version?


[DKMR22]: I. Diakonikolas, D. M. Kane, P. Manurangsi, and L. Ren, Cryptographic Hardness of Learning Halfspaces with Massart Noise

[Tie23]: S. Tiegel, Hardness of Agnostically Learning Halfspaces from Worst-Case Lattice Problems

**Questions:**

See last part of above section.

**Limitations:**

yes

---

> ### Author Rebuttal · Authors · 2024-08-05
>
> Thank you for your many helpful comments, and suggested references. We agree with all of your suggestions regarding the introduction and citations, and will fix them in a revision.
>
> We agree additional care can be taken to clarify the technical overview, emphasizing conceptual points and quantitative gains over formulas and full proofs.
>
> Lines 129-136: Re: the $\epsilon$ dependence, the number of iterations of both of our methods scales as $\frac 1 {\epsilon^2}$, but to implement each iteration of [CKMY20], they need to rejection sample from a region. The [CKMY20] termination condition only guarantees this region has at least $\epsilon$ probability mass, yielding a $\frac 1 \epsilon$ overhead. Re: the $\gamma$ dependence, this is similar to the following comment on cutting plane methods, which require certificates with much smaller failure probabilities than the first-order regret minimization approach we use. The [CKMY20] method is cutting plane-based, so their certificates are less sample efficient.
>
> Lines 153-162: The key difference is that cutting plane methods require high-probability guarantees on separating oracles being valid (as they are not robust to occasional errors); standard probability boosting techniques require $\gtrsim \frac 1 {\gamma^2}$ samples for sub-Gaussian concentration to kick in. On the other hand, the projected gradient regret minimization method we use is tolerant to any unbiased estimator with a second moment bound, so it only needs one sample per iteration, leading to a better $\frac 1 \gamma$ dependence.
>
> Lines 172-174: At the time of submission, we made the presentation decision to use a reweighting which adds $\gamma$ to the denominator rather than implementing the push-away operation (in the halfspace case), as it is conceptually simpler and adds less overhead to the proof. In light of the error we noted in the meta-comment, this distinction is no longer relevant, and in our revision we will include our new corrected proof for Massart GLMs.
>
> We plan to add discussion of these important points to the technical overview, and more generally include more comparisons to previous approaches, mentioning in greater detail why they incurred suboptimal sample complexities. In particular, we will spell out the bottlenecks to improving cutting plane methods (such as [CKMY20]'s approach) in more detail, as well as other previous approaches in the literature. We hope that this response was clarifying, and that our revision plan elevates our paper in your viewpoint.
>
> *[CKMY20] Sitan Chen, Frederic Koehler, Ankur Moitra and Morris Yau. Classification Under Misspecification: Halfspaces, Generalized Linear Models, and Connections to Evolvability. NeurIPS 2020*

---

> > ### Comment · Reviewer_kyaH · 2024-08-13
> >
> > Thank you for the detailed response! I think the above discussion makes the technical contribution of the submission over previous work much clearer. I have increased my score accordingly. I also appreciate (and acknowledge) the authors official rebuttal regarding the error and fix in the Massart GLM case.

---

### Official Review · Reviewer_TiaP · 2024-07-12

**Soundness:** 4
**Presentation:** 3
**Contribution:** 4
**Rating:** 7
**Confidence:** 3

**Summary:**

This paper focused on the fine-grained analysis of learning $\gamma$-margin halfspace with massart noise. The authors designed a new certificate vector $g$ by dividing the gradient vector of leaky ReLU by $|w^\top x| + \gamma$, and showed that when the hyperplane $h_w(x)$ has large 0-1 loss $\ell_{0,1}(w)\geq \eta + \epsilon$, the certificate vector aligns well with the direction of $w - w^*$, i.e., $g^\top(w. - w^*)\geq \epsilon$, hence a perceptron-like algorithm enjoys fast convergence to the optimal hyperplane $h_{w^*}(x)$. In the end, the authors showed that the proposed Perspectron algorithm requires only $O((\gamma\epsilon)^{-2})$ samples, matching with the sample complexity of learning margin halfspaces with RCN. The authors also extended the technique to massart GLM problem and achieved a similar $O((\gamma\epsilon)^{-2})$ sample complexity.

**Strengths:**

I enjoyed reading this paper due to its clarity and fluency in presentation. The authors did a good job explaining the intuitions and ideas. The result of this paper is very interesting to me, as I have always thought learning halfspaces with massart noises is much harder than with RCN, and this paper provided a surprising result that shows learning with massart noise can be achieved with similar sample complexity. The method (new certificate vector) that the authors proposed is also interesting, and can be informative for future research.

**Weaknesses:**

No serious weakness of this paper, but there are several typos here and there.
For example line 263 should be $w^*$, there should be an $x$ in the end of line 270, etc.

**Questions:**

I am not very familiar with massart GLMs. If I understand correctly, it seems the goal of massart GLM is still trying to find a classification hyperplane, but under this massart GLM model, the massart noise $\eta(x)$ is bounded by a function of $w^*\cdot x$. If so, isn't massart GLM a sub-class of massart model, as we have further constraints on $\eta(x)$ (in addition to simply restricting $\eta(x)\leq \eta$), hence perhaps implying that massart GLM is simpler than massart model?

**Limitations:**

The authors have addressed the limitations adequately.

---

> ### Author Rebuttal · Authors · 2024-08-05
>
> Thank you for your encouraging review, and we will make sure to address your typo catches. Re: your question about Massart GLMs, we will definitely provide more clarifying discussion on this point, as it is somewhat confusing. In particular, in a Massart GLM (following Definition 2), the optimal decision rule (evaluated by zero-one loss) is given by a halfspace, because the error rate is always $\le \frac 1 2$. However, the model is more general than Definition 1, because it allows a non-uniform maximum noise rate depending on $\mathbf{w} \cdot \mathbf{x}$ (in the halfspace model, the maximum noise rate is uniformly $\eta$). In summary, both of the following are true: Definition 1 is a special case of Definition 2, but the Bayes optimal prediction rule under Definition 2 is a halfspace (our algorithm also returns a halfspace).

---

> > ### Comment · Reviewer_TiaP · 2024-08-13
> >
> > I thank the authors for their response and corrections. I would like to remain my evaluation.

---

### Official Review · Reviewer_LLF4 · 2024-07-15

**Soundness:** 3
**Presentation:** 4
**Contribution:** 3
**Rating:** 6
**Confidence:** 3

**Summary:**

The paper considers the problem of learning halfspaces with a margin, under the Massart noise model. The Massart noise model generalizes the Random Classification Noise (RCN) Model: while in the RCN model, each label is flipped with fixed probability \eta, in the Massart noise model the the probability of flipping the label can be a function of the covariates bounded above by \eta.  The paper asks the question of whether one can design learning algorithms under the Massart noise model, matching the sample complexity under the RCN model, and answers it positively. Specifically, the paper proposes a proper learning algorithm, with sample complexity 1/(\epsilon^2 \gamma^2), matching the state of the art under the RCN model  (\epsilon is the error of the algorithm, \gamma is the margin). The results are also extended to the case of generalized linear models.

**Strengths:**

- The paper makes a concrete improvement to the sample complexity of learning halfspaces under Massart Noise, a classic problem in learning theory.

- The proposed algorithm is natural and simple, and the main ideas of the paper are explained well.

**Weaknesses:**

The set of people interested in the fine-grained complexity of learning under Massart Noise might not be very broad. So the significance of the results seems moderate.

**Questions:**

NA

**Limitations:**

Yes

---

> ### Author Rebuttal · Authors · 2024-08-05
>
> Thank you for your reviewing efforts; we are glad you found our algorithm natural and simple, and that it was explained well. We mention that beyond our main technical result, from a conceptual standpoint, our paper advances a line of work giving faster learning algorithms under realistic noise models, likely to be of broad general interest.
> We are optimistic that the insights of our paper may lead to follow-up developments, of simpler and more noise-robust algorithms in more general settings, e.g., learning noisy multi-index models (which includes fine-tuning a neural network as a case).

---

> > ### Comment · Reviewer_LLF4 · 2024-08-11
> >
> > I thank the authors for their response. After reading the response as well as other reviews, I am inclined to keep my original score.

---

### Author Rebuttal · Authors · 2024-08-05

We thank the reviewers for their positive feedback on our paper!

We would like to point out a technical issue in the current proof of Theorem 4 for learning **Massart GLMs**, which we found after the submission of our paper. The issue can be resolved through a concise extension of the proof of Theorem 3 (which we provide below), albeit with a worse sample and runtime complexity scaling as $\tilde{O}(1/(\epsilon^4 \gamma^2))$. This sample complexity is still an improvement to the comparable prior work [CKMY20] on Massart GLMs across all parameters, see discussion in Lines 107-115.

 **Our main statement and proof on Massart halfspaces (Theorem 3) is correct and remains unchanged.**

Despite the additional factor of $1/\epsilon^2$ in the sample complexity of learning Massart GLMs, the qualitative value of our result remains: a simple SGD-based algorithm simplifies and improves the best known algorithms for Massart GLMs with known activations.

Our revised approach for Massart GLMs is in line with our overall message, by showing that the Perspectron algorithm (Algorithm 1), after a slight parameter modification, achieves state-of-the-art guarantees for a more general noise model, in addition to halfspaces (Definition 1). This emphasizes the value of our simple algorithmic approach.


### Technical issue with current proof

Lemma 6, Part 2 is incorrect. It is only true when $\mathbf{w}$ is a unit vector. The issue stems from the fact that the added "Push-away" term is potentially unbounded when $\|\mathbf{w}\|$ is small. Intuitively, it is impossible to induce a large margin in an unnormalized small direction.

### Working Fix

As a consequence of this error, our analysis for Massart GLMs using the Push-away operation fails to go through. We propose a simple fix achieving a sample complexity of $\tilde{O}(1/(\gamma^2\epsilon^4))$. This still improves over prior work, but does not match our halfspace result.

We now present the details of our fix. Instead of the push-away operator, we use $|\mathbf{w}\cdot\mathbf{x}| \to |\mathbf{w}\cdot\mathbf{x}|+\frac{\gamma\epsilon}{2-\epsilon}$ as the modified denominator of the separating hyperplane in Lemma 5, thus bounding the norm of the step in our iterative method by $O(1/\epsilon \gamma)$. Combining this with the iterative method leads to the new sample complexity bound. We will add the complete proof in the final version and can also attach a pdf containing it if requested by the reviewers. We now present the proof of correctness for the new separating hyperplane, by highlighting the changes to the steps in the proof of Lemma 5.

*Lemma.* Let $D$ be an instance of $\sigma$-Massart GLM model with margin $\gamma$ and $\ell_{\text{0-1}}(\mathbf{w})\geq \text{opt}_{\text{RCN}}+\epsilon$. Then, we have that

 $\mathbf{E}_{(\mathbf{x},y)\sim D}[\frac{(\sigma(\mathbf{w}\cdot\mathbf{x})-y)}{|\mathbf{w}\cdot \mathbf{x}|+\alpha\gamma}\mathbf{x}]\cdot (\mathbf{w}-\mathbf{w}^*)\geq \epsilon$, where $\alpha=\frac{\epsilon}{2-\epsilon}$.

*Proof*. We borrow notation from the proof of Lemma 5. The only change from Lemma 5 is in how we lower bound $g(\mathbf{x})$. The analysis for the case $\mathbf{x}\in A$ is identical as it does not depend on the normalization used in the denominator of the separating hyperplane. We now analyse the case $\mathbf{x}\notin A$. There are two subcases.

First, suppose $|\mathbf{w}\cdot\mathbf{x}|\leq |\mathbf{w}^*\cdot \mathbf{x}|$. Let $c(\mathbf{x}):=\frac{|\mathbf{w}^*\cdot\mathbf{x}|}{|\mathbf{w}\cdot\mathbf{x}|}$. In this case, we have that $g(\mathbf{x})\geq\beta(\mathbf{x})\cdot\frac{|\mathbf{w}\cdot \mathbf{x}|+|\mathbf{w}^*\cdot\mathbf{x}|}{|\mathbf{w}\cdot\mathbf{x}|+\alpha\gamma}=\beta(\mathbf{x}) \cdot \frac{1+c(\mathbf{x})}{1+\alpha\frac{\gamma}{|\mathbf{w}^*\cdot\mathbf{x}|}c(\mathbf{x})}\geq \beta(\mathbf{x})\cdot\frac{1+c(\mathbf{x})}{1+\alpha c(\mathbf{x})}\geq \beta(\mathbf{x})(2-\epsilon)$, where the third inequality follows from $|\mathbf{w}^*\cdot\mathbf{x}|\geq \gamma$, and the final inequality follows from $\frac{1+c}{1+c\alpha}\geq 2-\epsilon$ for any $c\geq 1$ and $\alpha=\frac{\epsilon}{2-\epsilon}$. Thus, $g(\mathbf{x})\geq |\sigma(\mathbf{w}^*\cdot\mathbf{x})|+\beta(\mathbf{x})-\epsilon$.

Finally, consider the case where $|\mathbf{w}\cdot\mathbf{x}|\geq |\mathbf{w}^*\cdot \mathbf{x}|$. Here, we have that $g(\mathbf{x})=(|\sigma(\mathbf{w}\cdot\mathbf{x})|+\beta(\mathbf{x}))\cdot\frac{|\mathbf{w}\cdot\mathbf{x}|+|\mathbf{w}^*\cdot\mathbf{x}|}{|\mathbf{w}\cdot\mathbf{x}|+\epsilon\gamma}\geq |\sigma(\mathbf{w}^*\cdot\mathbf{x})|+\beta(\mathbf{x})$ as $|\mathbf{w}^{*}\cdot \mathbf{x}|\geq \gamma$.

Thus, we have proven that $g(\mathbf{x}) \ge |\sigma(\mathbf{w}^*\cdot\mathbf{x})|\pm \beta(\mathbf{x}) - \epsilon$ pointwise, where the $\pm$ depends on whether $x \in A$. We now repeat the steps from Lemma 5 to complete the proof, by comparing $g(\mathbf{x})$ to the zero-one error at $\mathbf{x}$, as done in Lemmas 1 and     2.

---

### Decision · Program_Chairs · 2024-09-25

**Decision:**

Accept (spotlight)

**Comment:**

This paper improves the state of the art sample complexity for learning halfspaces with massart noise, where the noise in each example is upper bounded by \eta. They match the \eta + \epsilon learning guarantee established in previous works [DGT19,CKMY20] with an improved sample complexity, and also give some extensions of their result to the more challenging GLM + massart noise setting studied by prior work. All of the reviewers agreed this advances the start of the art in this area with a simple and clean approach.